# Hydrodynamic and hydrological processes within a variety of coral reef lagoons: Field observations during 6 cyclonic seasons in New Caledonia

Oriane Bruyère[1], Benoit Soulard[1], Hugues Lemonnier[1], Thierry Laugier[1], Morgane Hubert[1], Sébastien Petton[2], Térence Desclaux[1], Simon Van Wynsberge[1], Eric Le Tesson[1], Jérôme Lefèvre[3], Franck Dumas[4], Jean-François Kayara[5], Emmanuel Bourassin[1], Noémie Lalau[1], Florence Antypas[1], Romain Le Gendre[1]

[1]Ifremer, UMR 9220 ENTROPIE (IRD, Univ. Réunion, IFREMER, Univ. Nouvelle-Calédonie, CNRS), BP 32078, 98897 Nouméa Cedex, New Caledonia
[2]Ifremer, UMR 6539 LEMAR, (Ifremer, Univ. Brest, CNRS, IRD), 11 Presqu'île du Vivier, F-29840 Argenton en Landunvez, France
[3]IRD, UMR 9220 ENTROPIE (IRD, Univ. Réunion, IFREMER, Univ. Nouvelle-Calédonie, CNRS), BPA5, 98948 Noumea, New Caledonia
[4]SHOM/STM/REC, 13 Rue de Châtellier CS 92803, 29228 Brest CEDEX 2, France
[5]Direction du Développement Economique et de l'Environnement, Service des milieux et ressources aquatiques, 98859 Koné, New Caledonia

Correspondence to: Romain LE GENDRE (romain.le.gendre@ifremer.fr)

**Abstract.**

From 2014 to 2021 during the cyclone seasons, extensive monitoring of the hydrodynamics within a variety of lagoons of New Caledonia was conducted as a part of the PRESENCE project (PRESsures on coral Ecosystems of New CalEdonia). The PRESENCE project is aimed at building an efficient representation of the land-lagoon-ocean continuum at Grande Terre, New Caledonia's main island. Overall, coastal physical observations encompassed five different lagoons (four of which were never before monitored) and at least eight major atmospheric events ranging from tropical depressions to category 4 cyclones. The main objectives of this study were to characterize the processes controlling the hydrodynamics and hydrology of these lagoons (e.g., ocean-lagoon exchanges, circulation, level dynamics, temperature and salinity variability), and to capture the magnitude of change during extreme events. An additional objective was to compile an adequate data set for future use in high-resolution hydrodynamic models. Autonomous oceanographic instruments were moored at strategic locations to collect time series of temperature, salinity, pressure and eulerian currents. Additionally, lagrangian surface currents were observed through deploying drifter buoys, and cross-shore hydrological profile radials were carried out using CTDs. In total, five survey campaigns were conducted, beginning with the SPHYNX campaign which lasted 15 months (December 2014 to February 2016) in the Hienghène-Touho lagoon and ended with the 9 months NEMO campaign (September 2020 to April 2021) in Moindou lagoon. Between these were the 5 months NOUMEA campaign (December 2016 to April 2017) in Noumea lagoon, the 6 months ELADE campaign (February to August 2018) in the Poe lagoon, and the 5 months CADHYAK campaign

(December 2019 to May 2020) in Koumac lagoon. In addition to characterizing these lagoons, the data set identifies important features and processes, such as the presence of internal waves on forereefs, wave-driven fluxes over reef barriers and exchanges through passes. Signatures from strong events were also identified, including surges, thermal drops inside lagoons, and massive flash flood plume dispersion. Raw data sets were processed, controlled for quality, validated and analyzed. Processed files are made publicly available in dedicated repositories on the SEANOE marine data platform in NetCDF format. Links (DOI) of individual data sets are provided herein.

## 1 Introduction

In addition to harboring an estimated 70% of coastal marine diversity coral reefs provide a large range of benefits and services for millions of humans (Hughes et al., 2017), including, being a key source for seafood and providing coastal protection from oceanic forcings. Despite their importance coral reef habitats and abundances are declining drastically, predominantly due to increased exposure to a range of climatic and local stressors (França et al., 2020). Examples of threats to coral reef ecosystems include thermal-stress (Lough et al., 2018) which may induce disturbances in coral functioning (bleaching), tropical storms or cyclones which may cause mechanical damage (Cheal et al., 2017), and flash flooding which alters water quality and incidentally may induce disturbances in coral functioning (Tan et al., 2012; Desclaux et al., 2018). Additionally, threats may be enhanced during periods of extreme events, such as tropical cyclone seasons, or can be controlled or mitigated through various hydrodynamic processes. Coastal hydrodynamics play a central role by controlling distribution, growth and resilience of coral communities (Lenihan et al., 2015; Rogers et al., 2016; Shedrawi et al., 2017). In turn, coastal hydrodynamics are controlled by geomorphological aspects (e.g., bottom topography, mean depth, degree of openness) which influence circulation and thermodynamic major features such as water exchange rate, renewal time, mixing, and heat budget (Umgiesser et al., 2014) resulting in lagoons exhibiting a wide range of responses to extreme events.

New Caledonia (NC) offers a high diversity of coral reef complexes covering more than 4500 km$^2$ (Andréfouët et al., 2009), and spread across a variety of lagoons of contrasting sizes and shapes. Here, we refer to "lagoon" as the water body located between the shore and the reef crest. In the past research works and knowledge on lagoon scale hydrodynamics (either from observations or modelling) were limited to two lagoons located on the south-western side of the main island (Grande Terre); the large South-West lagoon (SW) in front of the capital Nouméa (Douillet, 1998; Ouillon et al., 2010; Jouon et al., 2006), and the Ouano lagoon which is located approximately 80 km northwest of Nouméa (Sous et al., 2017; Chevalier et al., 2015). Furthermore, to our knowledge, none of the observational strategies where specifically dedicated to observe the high-frequency signature of cyclonic events on hydrodynamics and hydrology of the Grande Terre lagoons.

In this context, the PRESENCE project (PRESsures on coral Ecosystems of New CalEdonia), funded by New Caledonia institutions (Government, North and South Province) has been launched to partially fill this gap. The aim was to provide a

synoptic view of lagoon functioning using field measurements, satellite observations and high-resolution hydrodynamic modelling to characterize inter-lagoon heterogeneity across lagoons of Grande Terre. This project focused on physical processes occurring in coastal ecosystems and especially on the interactions and exchanges along the land-lagoon-ocean continuum. In this framework, field observations were undertaken in four unmonitored lagoons and one previously monitored lagoon, with a focus on austral summer i.e., cyclone season, to identify mean circulation and forcings, as well as to highlight the scales of variability.

Data presented in this paper contributes to the knowledge and understanding of the array of physical processes at play within New Caledonian lagoons. It represents an essential database to improve realism of numerical model development and experiments. Five main surveys were conducted around Grande Terre lagoons in the 2014-2021 period, covering 6 consecutive cyclonic seasons. After a brief introduction of the lagoon hydrodynamics in New Caledonia lagoons in Section 2, study sites and observational strategies are then described in Section 3. Sensors types, processing and quality control methods are detailed in Section 4. Finally, general outputs of observations acquired are described in Section 5.

## 2 New Caledonia Lagoon's context

The New Caledonian archipelago is located in the south-west of the South Pacific Ocean between 19 °S - 23 °S latitude and 163 °E - 167 °E longitude. The main island, called Grande Terre, is an elongated mountainous island (~ 400 km long and ~ 50 km wide) oriented southeast to northwest. Grande Terre is surrounded by the world's second longest semi-continuous coral reef barrier system (1 744 km$^2$), after the Australian Great Barrier Reef. The system includes numerous deep channels that allows for water exchange with the open ocean. The barrier reef system and coastline of Grande Terre delimit a vast number and variety of lagoon seascapes which jointly cover 21 896 km$^2$ (Andréfouët et al., 2009). Since July 2008, four marine areas of the main island were listed under the United Nations Educational, Scientific and Cultural Organization (UNESCO) as World Heritage Sites, which consequently promoted monitoring, management, and conservation actions to maintain the integrity of the reef-lagoon ecosystems.

Grande Terre lagoons are made up of highly diverse geomorphologies and coral reef complexes (Andréfouët et al., 2009). While distance from shore to barrier reef ranges from ~ 2 to ~ 30 km, depth varies between ~ 1 and ~ 50 m inside lagoons and drop rapidly to over 600 m along the fore-slope of the barrier reef (Fuchs et al., 2013). The western coastline of Grande Terre is characterized by shallow and narrow lagoons, while along the eastern coastline lagoons are generally deeper and wider. Amongst the lagoons of the western coast, the north-west and south-west lagoons are wider and deeper contrary to the central west area where lagoons are skewed towards land forming fringing reef systems in places. Along the eastern coast the barrier reef system is also partly drowned and sparser, offering more openness to the open ocean. Extending from the north western

tip on Grande Terre is the "Grand Lagon Nord", New Caledonia's largest lagoon which extends ~ 140 km northward into the South Pacific Ocean.

## 2.1 Atmospheric forcings

New Caledonia's weather displays the typical features of central-western pacific tropical climate. That is, seasonality and inter-annual variability are mainly driven by the position of the South Pacific Convergence Zone (SPCZ) and the El Niño/Southern Oscillation (ENSO) respectively, although other timescales of modulation are induced by large-scale fluctuations such as Madden-Julian and Interdecadal Pacific oscillations (Dutheil et al., 2021). The south-east trade winds, which represent 70% of the yearly wind occurrence within region are prevalent all year long although slightly weaker and less constant during the cold season. Moreover, the intensity of winds is defined more by diurnal rather than seasonal variability (Caudmont and Maitrepierre, 2006). In phase with the seasonal migration of the SPCZ, rainfall is less abundant in austral winter (May–October) than in summer (November–April) with maximum precipitation observed between February and March (Payri et al., 2019). Due to the mountainous shape of the Grande Terre (the two highest peaks Panié and Humboldt are over 1600 m), orographic effects induce modulation of the SE trade winds regional regime resulting in more precipitation on the windward eastern coast and eventually leads to higher freshwater input into to the lagoons of the eastern coastline (Lefèvre et al., 2010; Terry and Wotling, 2011).

The ENSO impacts precipitation dispersion over the whole Pacific Basin and depending on its phase, leads to strong inter-annual variability of rainfall over New Caledonia. In its positive phase (El Niño), a ~ 20–50 % decrease in rainfall can be observed over New Caledonia, while during La Niña an up to 50 % increase in precipitation can be observed (Nicet and Delcroix, 2000; Moron et al., 2016).

During austral summers, tropical cyclones form or move into the New Caledonia basin with an average occurrence of 1.5 per year bringing extreme winds and rainfall (Grenz et al., 2013). During La Niña events, tropical cyclones are more frequently observed between New Caledonia and Vanuatu (Dowdy et al., 2012).

## 2.2 Oceanic forcings

The regional subsurface circulation around New Caledonia is characterized by two large ocean currents called the North Caledonian Jet (NCJ) and the South Caledonian Jet (SCJ). They both flow westward and are ramifications of the South Equatorial Current (SEC) which bifurcates into NCJ and SCJ branch before reaching Loyalty Islands (Couvelard et al., 2008; Marchesiello et al., 2010). In the vicinity of the Grande Terre and Loyalty Islands, Gasparin et al., 2011 described a subpart of the NCJ called the East Caledonian Current (ECC) as a boundary current located between 10 and 100 km from the east coast of New Caledonia and flowing north-westward. The alongshore transport around the main island is predominantly composed of two currents flowing southeastward, i.e., against the mean wind direction, in the subsurface layer: along the western coast

is the more persistent Alis current, and along the eastern coastline is the more variable Vauban current located in the Loyalty Channel, i.e., between the Loyalty Islands and Grande Terre (Cravatte et al., 2015). The final major current influencing the regime around the main island is the Sub Tropical Counter Current (STCC), a subbranch of the original East Australian Current, which is flowing eastward from Australia and arrives along the southern coast of Grande Terre.

Oceanic Sea Surface Temperature generally ranges from 23-24 °C during winter to over 28 °C during the summer season
(Payri et al., 2019). Furthermore, within latitudes, oceanic surface waters are cooler on the west side on the Grande Terre as opposed to the east side. The orientation of the main island relative to the trade winds generates favorable conditions for wind-driven upwelling along the west coast and downwelling along the east coast (Marchesiello et al., 2010). Intense cooling events during summer season (October to March) were highlighted for the first time by Hénin and Cresswell, 2005 on the south-western outer reefs of NC. Sea Surface Temperature from satellite images observation associated with wind trade events allow
to describe a cooling of surface temperature from 2-4 °C in this area. Few cooling events occur on the east coast especially during westerly and north-westerly winds (Hénin and Cresswell, 2005). Upwelling triggering processes and consequences have been thoroughly studied, either by means of modelling (Alory et al., 2006; Marchesiello et al., 2010; Fuchs et al., 2013) or using observations (Cravatte et al., 2015; Ganachaud et al., 2010; Neveux et al., 2010).

Considering the wave climate around New Caledonia, the only significant literature to our knowledge is study by Ouillon et
al., 2010 who describe using literature the seasonality of waves around the SW lagoon. The highest swell generally comes from the south-south-east between March and May (mean monthly value range: 2.3-2.4 m) and the lowest swell between October and January (around 1.9–2.0 m). The yearly mean wave period around this south-western part of the Grande Terre ranges from 7.1 s to 8.7 s.

**2.3 The ocean lagoon interface**

Coral reef barrier systems, act as an efficient buffering mechanism, protecting coastlines and lagoons from oceanic forcings such as huge oceanic states, tsunamis, and tropical cyclones, which in turn reduce coastal inundation and erosion. Despite the physical barrier, the ocean-lagoon interface allows oceanic waters to pass into lagoons through swell and wave breaking, tidal currents, which can be a particularly important feature for shallow lagoon circulation. Within Grande Terre, the ocean-lagoon
interface has been partially considered by Bonneton et al., 2007 and Sous et al., 2019. The former describes the tidally wave-induced currents along the Aboré reef barrier in front of Nouméa (South-West lagoon of Grande Terre) using an analytical model, while the latter conducted field and numerical experiments to describe wave transformation over the barrier reef within the Ouano reef-lagoon system The importance of the cross-reef fluxes (due to incident waves breaking and tide) across barrier reef interfaces for efficient circulation and distribution of water properties has been highlighted by Chevalier et al., 2015 and
Sous et al., 2017 at Ouano lagoon. In Sous et al., 2020, the authors go on to study more precisely the different terms in the

momentum balance over the Ouano barrier reef and describe for the first time in Grande Terre the regime shifts associated with calm hydrodynamic conditions and huge incident waves.

## 2.4 Lagoon dynamics

Tidal dynamics in NC was first studied by Douillet (1998) in the SW lagoon of using observational experiments coupled with a 2D numerical model. Within this lagoon, the dynamics were found to be driven mainly by semi-diurnal tides, especially when considering the M2 (lunar component) and S2 (solar component). The tidal range within the SW lagoon ranges from as low as 0.6 m during neap tides to as high as 1.4 m during spring tides (Douillet, 1998). While these cyclic tidal dynamics have remained constant, tidal forcing is episodically modified by extreme events such as tropical cyclones. Numerical experiments by Jullien et al., 2017 to study the effect of atmospheric surge and wave-setup during TC Cook which hit the east coast of NC in 2017, revealed atmospherically driven surge of up to 0.5 m and an anomaly due to wave-setup up to 0.25 m depending on the lagoon's geomorphologies.

In the SW lagoon, tidally generated currents are mainly aligned along the lagoon axis (except near passages) and can reach 0.2 m s$^{-1}$ for M2 components, and ~ 0.03 m s$^{-1}$ for S2 components (Douillet, 1998) while mean tidal currents are between 0.05 and 0.1 ms$^{-1}$ within the lagoon and approx. 0.2 - 0.3 m s$^{-1}$ in the passages (Ouillon et al., 2010). Observations using Acoustic Doppler Current Profilers (ADCP) at four different stations of Ouano lagoon showed a small vertical variability of currents (< 0.05 m s$^{-1}$) and a high variability depending on the mooring site, up to 0.7 m s$^{-1}$ observed in a passage (Chevalier et al., 2015). Finally, fields observations and numerical modelling were used conjointly to characterize circulation patterns and residence times in both SW lagoon (Jouon et al., 2006) and Ouano lagoon (Sous et al., 2017)

The spatial distribution and variation of temperature, salinity and water turbidity within the lagoons of Grande Terre are highly dependent on seasonality and are amplified during ENSO phases (Ouillon et al., 2005). During summer, bays and coastal areas record higher water temperature than lagoons while the contrary is true in winter. During El Niño, due to low freshwater inputs and enhanced evaporation, an increase in temperature and salinity is observed within lagoons while the opposite is observed during La Nina. Finally, the fate of rivers plumes (from the Dumbéa, Coulée, Pirogues rivers) and their consequences on the SW lagoon were studied through biogeochemical and sedimentological studies conducted by Pinazo et al., 2004; Ouillon et al., 2004; Drouzy et al., 2019. Together, these studies present the gradients of suspended particulate matter, nutrients and chlorophyll-a within the SW lagoon and highlight that the small bays display much higher values as opposed to the rest of the lagoon.

Generally, waves from ocean are strongly attenuated (wave breaking) by the barrier reefs and are minor contributors to circulation within the lagoons around Grande Terre, with their greatest presence being around reef passages which act as an interface where waves can enter into the lagoons. The SW lagoon however is dominated by high-frequency wind waves (Jouon et al., 2009). This is due to the SW lagoon being large enough to allow the trade winds to generate sufficient fetch to form

wind driven waves. Further observational studies by Aucan et al., 2017 in the SW lagoon define three main modes of wave-based circulation, namely: high frequency wind waves (period: 3-8 s), low frequency generated by incoming swell within the passages (period: 8–25 s) and infragravity waves (period: 25–400 s). Maximum significant wave height within the SW lagoon is observed to be between 1.0 m and 1.5 m (Jouon et al., 2009; Aucan et al., 2017).

## 2.5 Past and on-going observational strategies

From the hydrodynamic characteristics described above, one can see that field observations have mainly focused on the functioning circulation and exchanges within the SW lagoon (during CAMELIA unit research lifetime) and the Ouano lagoon with dedicated cruises (the R/V Alis). To our knowledge, data sets associated with these past studies are neither freely not easily accessible, A further observation network, called Reeftemps (Varillon et al., 2021), targets mainly subsurface temperature through long-term deployment of temperature loggers within coastal waters to evaluate the effect of climate change on coral reefs ecosystems in 14 countries and territories in the Pacific. Data sets are freely accessible on their web portal (*www.reeftemps.science*) and sporadically include other data sets acquired during related field campaigns. In New Caledonia, this data set only recently extended to other physical parameters (e.g., salinity and pressure) but remains restricted to stations distributed in and around the SW lagoon.

## 3    Study sites and observational strategies

### 3.1 Study overview

Between December 2014 and April 2021, five distinctive lagoons (Hienghène-Touho, Nouméa, Poe, Koumac, Moindou) around Grande Terre were outfitted with various instruments. The main goal of this observational strategy was to collect a robust data set to enable an improved understanding of hydrodynamics in a variety of unmonitored lagoons around Grande Terre during periods encompassing the cyclone seasons (from December to May). At least 8 major meteorological events ranging from a Tropical Depression (TD) to a category 4 cyclone (Fig. 1) were captured, as well as periods of contrasting Southern Oscillation Index (SOI). The length of data acquisition for each lagoon ranged from 2 to 15 months, and number of instruments moored for each campaign differed slightly depending on availability of funds instruments. These periods of data acquisition are defined as campaigns and are described in detail in section 3.2. Due to the necessity of frequent maintenance (e.g., due to fouling, battery replacement and data retrieval), most of surveys were delimited in legs lasting 3 months (except when retrieval was not possible (e.g., cyclonic conditions or COVID-19 induced travel restrictions).

Sampling strategies were designed to consider important features driving lagoon circulation and hydrology within each lagoon studied (refer to supplementary Figures: Fig. A1, Fig. B1, Fig. C1, Fig. D1) for a depiction of sampling strategies. The observations were obtained from long-term, moored instruments such as ADCP's and temperature profilers or from mobile

instruments such as CTDs and lagrangian surface drifters (cf. Section 4). Most moored instruments were deployed or recovered by scuba-diving.

Three primary scientific objectives were considered as a part of the sampling strategy, namely: (i) to characterize oceanic forcings acting on the lagoon, (ii) to quantify exchanges at lagoon-ocean interface, and (iii) to capture hydrodynamic signatures

of high energic events in the lagoon. For objective one, temperature and sea level were monitored on along the forereef adjacent to each lagoon studied to acquire a high-frequency sea level time series (for understating aspects such as incident waves breaking on barrier reefs, and surge dynamics), as well as thermal dynamics (for understanding aspects such as upwelling or internal waves). Objective two was dedicated at quantifying fluxes across the barrier reef as well as currents inside passes. Finally, objective three involved monitoring the hydrodynamics for a prolonged period over the summer season to allow for

capturing the unique signatures in lagoons associated with trade winds conditions as well as during high energy events (surges, plume dispersal, intensification of ocean-lagoon exchanges, etc.). occurring between the land-lagoon and lagoon-ocean interfaces.

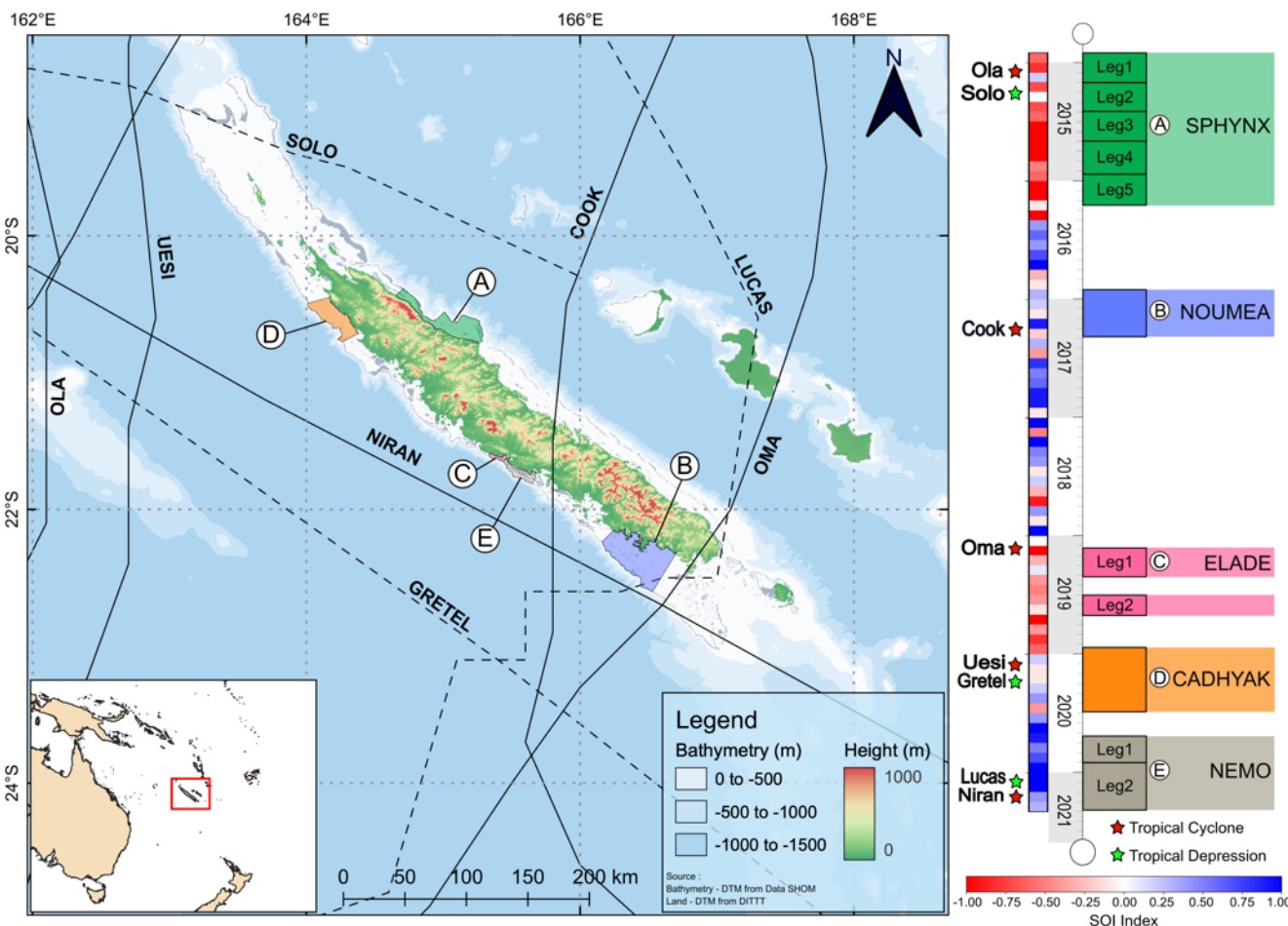

**Figure 1: Map of New Caledonia including the location of the five studied lagoons: (A) SPHYNX, (B) NOUMEA, (C) ELADE, (D) CADHYAK, and (E) NEMO. Dashed lines and solid lines represent paths of tropical depressions and tropical cyclones respectively. Chronology of the campaigns relative to the Southern Oscillation Index (SOI; blue: positive index, red: negative index) and the 8 major atmospheric events (red stars: Tropical Cyclones, green stars: Tropical Depressions) is depicted to the right of the map.**

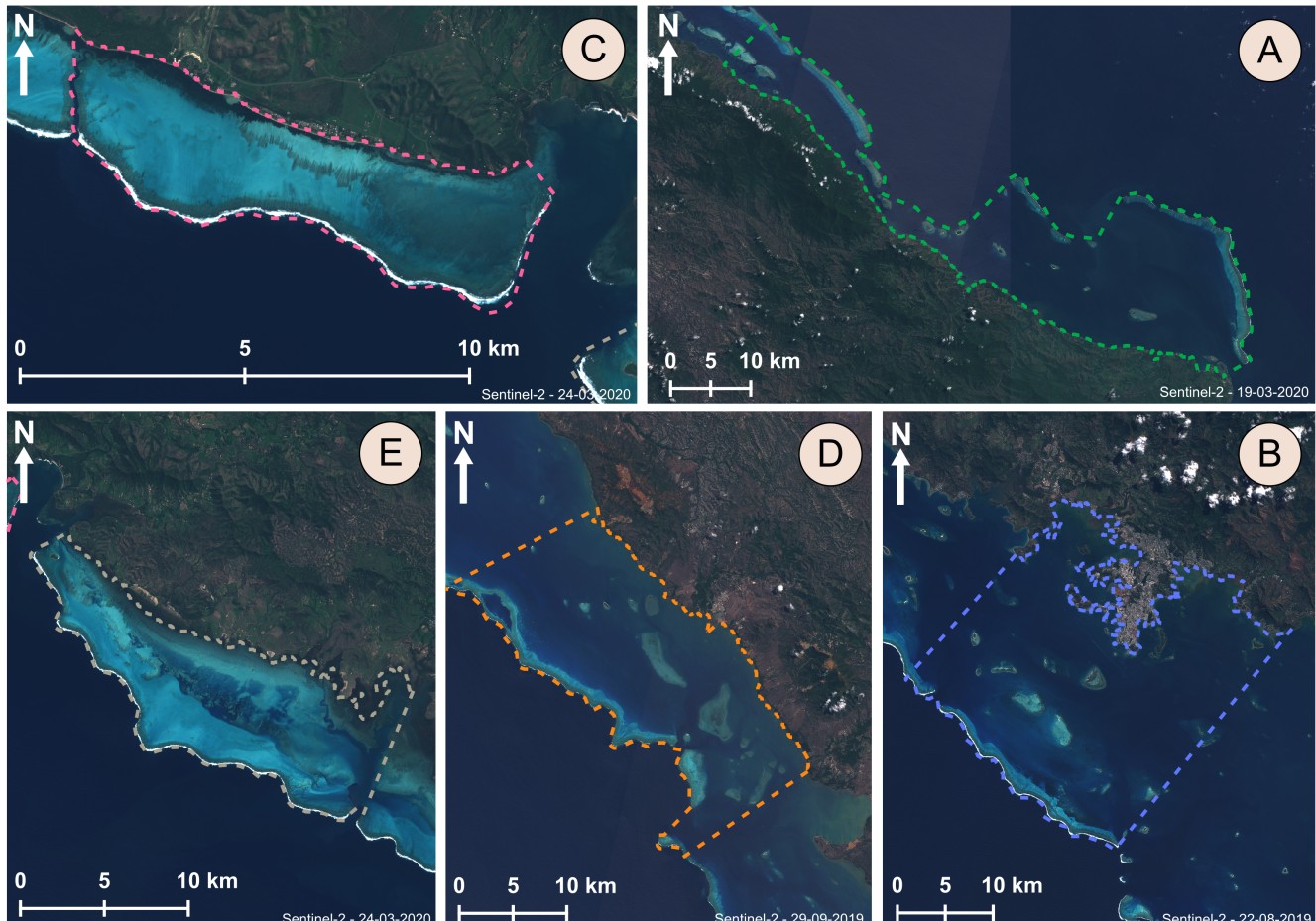

**Figure 2: Geomorphologies of the lagoons from Sentinel-2 imagery from European Space Agency (ESA): (A) Hienghène-Touho lagoon (B) SW lagoon (Nouméa) (C) Poé lagoon (D) Koumac lagoon (E) Moindou lagoon.**

### 3.2 The five campaigns and study sites

In chronological order, the first and longest campaign of the PRESENCE project was the SPHYNX campaign within the Hienghène-Touho lagoon. In total the data set covers a total period of 15 months spread over 5 Legs conducted between December 2014 and March 2016. This lagoon is part of the North-East Coastal Region classified among UNESCO world heritage sites. The lagoon of interest has an average depth of 29 m and its widest point stretches ~19 km between the coastline and reef crest. The lagoon is separated from the ocean with a camelback-shaped barrier reef punctuated with large passages

(Fig. 2). A chain of 6 islets surrounded by large reef flats is scattered parallel to the shore centered in the middle of the lagoon. The SPHYNX data set is mainly characterized by a negative SOI and includes 2 major atmospheric events, the TC "Ola" which brought with it a cumulative rainfall of around 50 to 100 mm in 48 h on the east coast and the TD "Solo" generating 100 mm of rain within 24 h along the east coast (source Météo France Nouvelle-Calédonie). In terms of human induced pressures, this lagoon is considered as "pristine or natural area" as only 4800 inhabitants live within the vicinity in tribe-based

village settings (source ISEE (Institut de la statistique et des études économiques), 2019). At the time of writing there are no mining activities or other major land use industries being conducted the upstream watersheds.

The NOUMEA survey was performed in a single leg in the south-western lagoon over a span of 3 months during 2016-2017 cyclonic season. The semi-opened lagoon has a funnel-shaped, seaward-extending geomorphology and on average measures

17 km between the coastline and barrier reef, and has a mean depth ranging between 15 and 20 m (Douillet, 1998). This lagoon undergoes several anthropogenic pressures due to the presence of the urban and peri-urban areas on and around Grand Nouméa (180 000 inhabitants). The NOUMEA data set is characterized by an SOI index fluctuation monthly between La Niña and El Niño phases and includes a single large atmospheric event, the TC "Cook" (a category 4 tropical cyclone). TC Cook was the most intense high energy event recorded during acquisition period for data presented in this paper, and produced winds with

average wind-speeds of 130 km h$^{-1}$, as well as heavy rainfall on the south-west Grande Terre.

ELADE observations was conducted in the Poé lagoon and aimed at understanding hydrodynamics and nutrients pathways that triggered a wash-up of green algae onto the shoreline in 2018 (Brisset et al., 2021). This UNESCO world heritage site is a key recreational area for the inhabitants of Nouméa, offering picnic, camping areas, and aquatic activities. In this area, the

270 barrier reef complex lies directly in front of the shore, i.e., a fringing reef, forming a shallow (< 4 m) lagoon of with a surface area of circa 25 km$^2$ and includes sandy terraces, patch reefs, as well as the reef crest and forereef structures. The lagoon is separated from the ocean with a continuous reef crest which lies approximately 2 km from coastline, only segmented by a deep (~25 m), narrow (< 200 m) pass called "Shark fault" (Amrari et al., 2021). The survey was performed along 2 separates legs in 2019 over summer and winter season respectively. The legs are characterized by a couple of months in the El Niño phase

and by the TC "Oma" which has impacted the Grande Terre with strong winds (> 100 km h$^{-1}$) and huge sea states.

During CADHYAK survey, observations were carried out over 2019-2020 cyclone season in the Koumac lagoon which is located on the north-western coast of the main island. This area is distinct from the others sites due to extensive mining and mining related activities within the vicinity of the Koumac lagoon. The lagoon is characterized by a barrier reef located approximately 12 km from the coastline and a depth ranging from 0 to 20 m. During the course of sensor deployment, two extreme meteorological events successively hit the Grande Terre, namely TC "Uesi" and TD "Gretel". The former generated 575.8 mm of rainfall (Koné station; Météo France Nouvelle-Calédonie) spread over three days and gusts of over 100 km h$^{-1}$ in Koumac, the latter, which occurred within a month generated 150 mm of precipitation in 48 hours.

The last field and most recent campaign under the PRESENCE project, was the NEMO campaign and was conducted in the Moindou lagoon during the 2020-2021 summer season. Observations covered a period of 8 months spread over two legs. The Moindou lagoon is characterized by a funnel-shaped geomorphology; it is narrowest, 2 km from coast to barrier reef at its northern most point and becomes wider further southward (~ 7 km near Moindou passage). The lagoon is mainly shallow (< 6 m) but includes some reticulated "deep holes". The surrounding semi-continuous reef barrier includes two shallow passes on the reef crest (Fig. 3). The NEMO data set was recorded during a moderate La Niña event and two significant atmospheric events happened: the strong TD "Lucas" in February and the severe TC "Niran" on March causing extreme gust of winds (189 km h$^{-1}$ at Nessadiou station; Météo France Nouvelle-Calédonie).

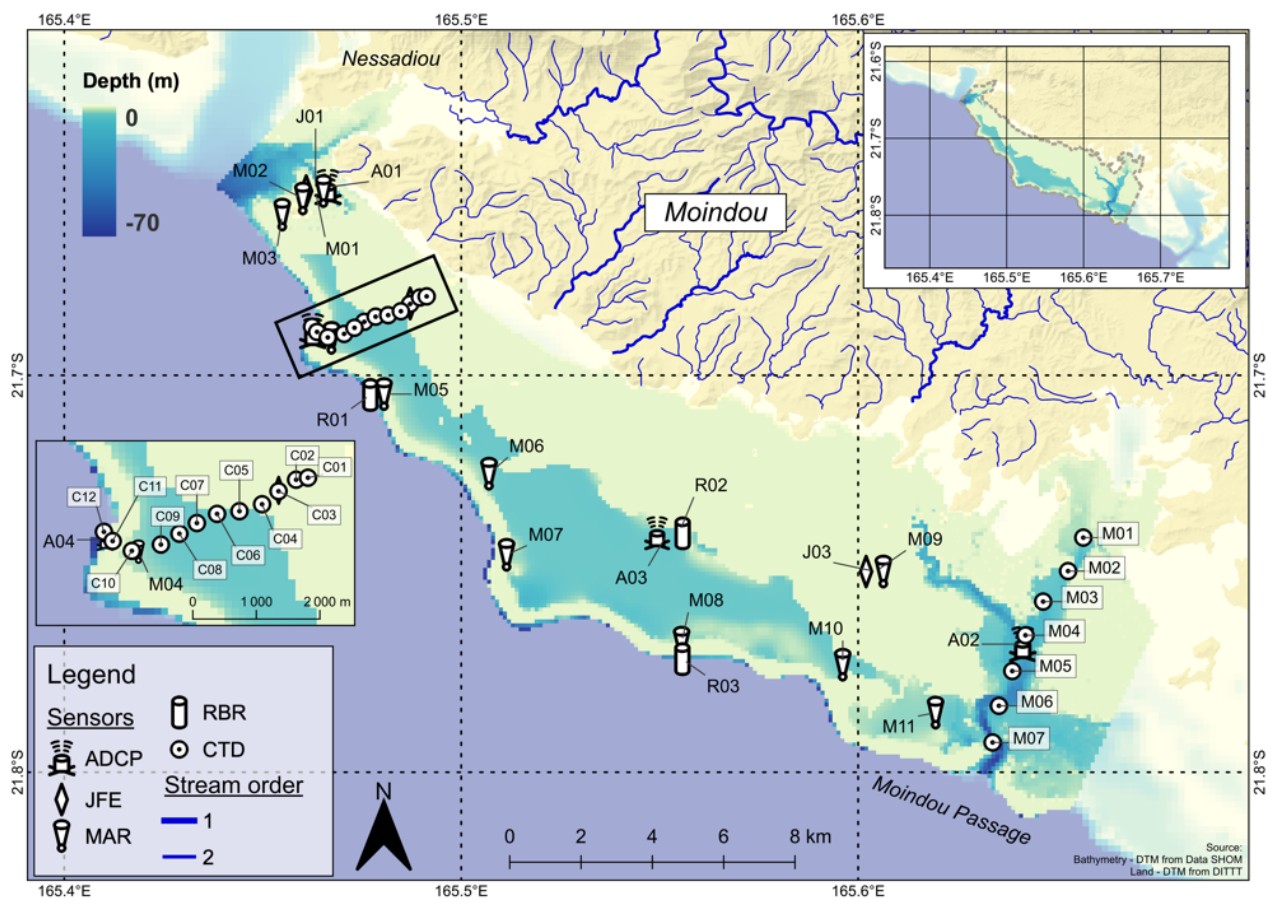

**Figure 3: A depiction of the observational strategy deployed within the Moindou lagoon during NEMO campaign. ADCP: Acoustic Doppler Current profiler; JFE: INFINITY-CTW; MAR: Marotte HS; RBR: RBRduo T.D or RBRduet T.D; CTD: Conductivity – Temperature – Depth.**

### 3.3 Companion data sets

Lagrangian drifters were deployed during additional field experiments encompassing the 2019-2020 cyclone season. All deployments were within the SW lagoon, specifically around the capital Nouméa. The drifters were deployed for 2 key projects, the first involved deployments around the Dumbéa bay as a part of the SEARSE (Imprints of river and estuary waters) project (Lemonnier et al., 2020), which aimed to investigate the influence of flash flood plumes dispersal on microbial communities. The second set of drifter releases, which lasted for 2 to 3 days, were specifically designed to calibrate numerical models developed for SAR (Search And Rescue) experiments and covered more broadly the waters around Nouméa.

## 4 Instruments, methods, and deployments

A variety of physical oceanographic instruments were deployed during the PRESENCE project. They include moored loggers dedicated either to temperature, salinity or pressure, or to eulerian currents (either punctual or profilers), and mobile loggers such as drifters (for characterizing surface lagrangian circulation), and Conductivity-Temperature-Depth (CTD) profilers capturing the vertical structure of the water column. A detailed list, containing all instruments used, their make and model, logging intervals and frequency used, and deployment location is provided within the supplementary Table A1. For instruments

fixed on the seabed, moorings were adapted to the habitats and locations in terms of weight and size to ensure stability even during high energetic episodes. Supports were thus made up of different materials ranging from concrete blocks, brake drums, or steel bars and were equipped with anodes to prevent corrosion or damage through electrolysis. To prevent biological fouling cayenne pepper and grease was spread on the body of the ADCP's and subsequently covered up with electrical tape. Similarly, to protect them from biofouling and mechanical damage, smaller compact loggers were deployed inside of PCV cylinders with

holes drilled in to allow for circulation of water. All the moored instruments were autonomous sensors and contained internal batteries and memory.

### 4.1 Compact loggers: Temperature, Salinity and Pressure

Three types of compact loggers were utilized across the various campaigns to monitor temperature, salinity and pressure. The first set of sensors used were SBE56 (SEA-BIRD Electronics Inc.; *https://www.seabird.com/sbe-56-temperature-*

320 *sensor/product?id=54627897760*) temperature loggers. The SBE56 allowed for recording fast and highly-accurate recording of temperature, and was moored at depths up to 26 m depending on station. The SBE56 used, were set to record temperature with a raw sampling frequency of 30 s.

The second set of loggers used were INFINITY-CTW (JFE Advantech Co., Ltd; *https://www.jfe-advantech.co.jp/eng/products/ocean-infinity.html*) temperature and conductivity loggers (Fig. 4.E). These instruments are

325 designed for long-term observations as they include a biowiper which cleans periodically the conductivity cell and prevents drifts in measurements. They were typically deployed using bursts of 10 samples at 1 Hz every 10 or 15 min. As one of the main objectives of these sensors was to capture plume dynamics, they were thus all moored in shallow water ranging from 2 to 5 m.

The third set of compact loggers used were two models of RBR Ltd. namely, RBRduo T.D, and RBRduet T.D (*https://rbr-*

330 *global.com/*). The RBR loggers were used to record temperature and pressure over the duration of campaigns (Fig. 4.D). These loggers were moored predominantly along the external slopes at depths of 12 m and set to sample at 1 Hz intervals to provide high-frequency pressure data and offer the possibility to derive sea-state parameters such as incident waves. When deployed within the lagoons they configured to a lower sampling frequency (10s) to allow long deployments and thus capture mainly tidal and surge signals.

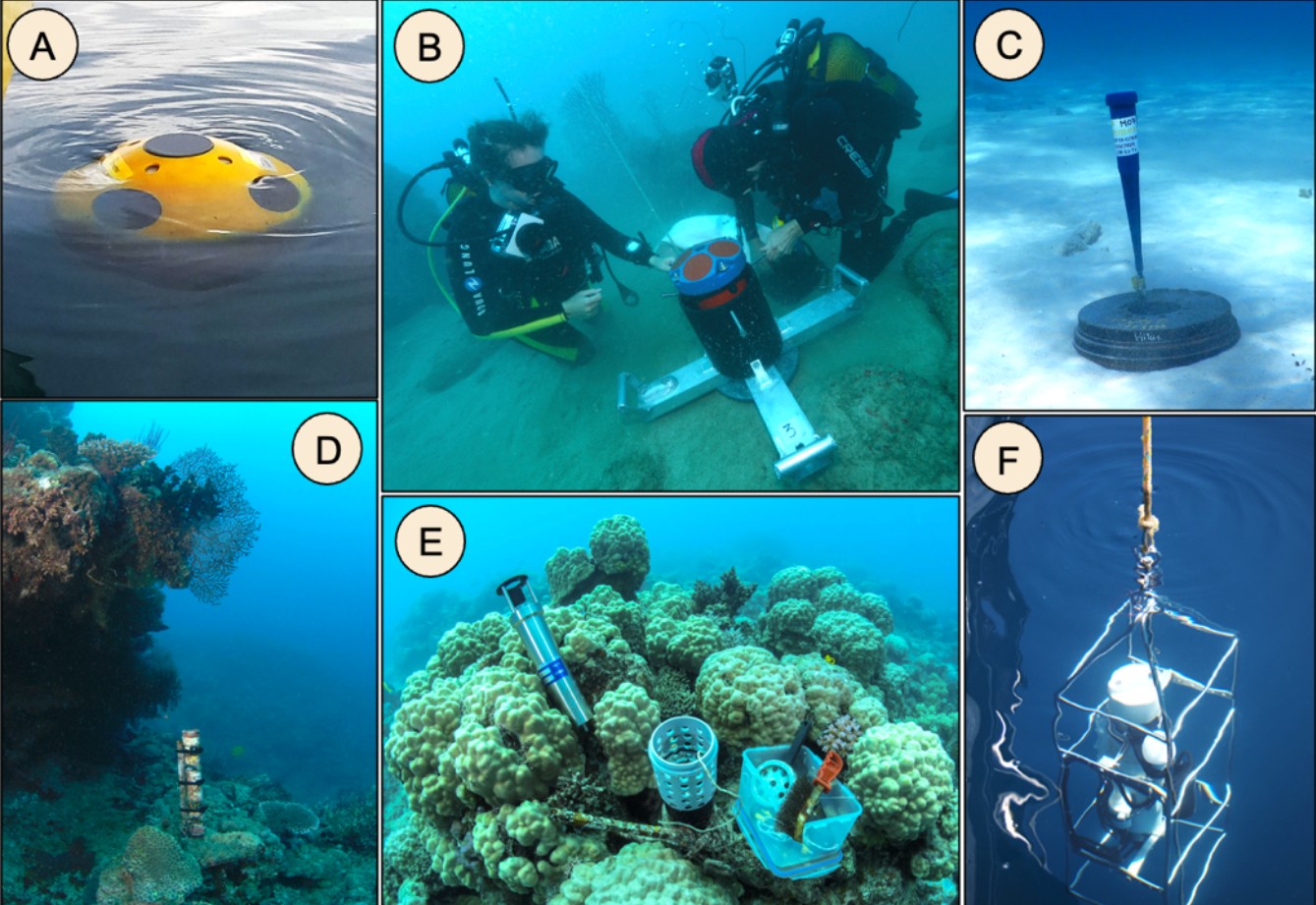

**Figure 4: Photographs of deployed instruments for PRESENCE project. (A) Reef drifter (B) Acoustic Doppler Currents Profilers (C) Current meter Marotte HS (D) RBRduet T.D (E) Maintenance of a JFE Advantech logger (F) CTD SBE 19plus profiler. Credit photo: IFREMER LEAD-NC; US IMAGO.**

**4.2 Eulerian and lagrangian current measurements**

Three types of instruments namely, drag-tilt current meters, ADCPs and reef drifters, were utilized across the various campaigns to measure eualrian and lagrangian currents, and included drag-tilt current meters. All drag-tilt current meters were Marotte HS (Marine Geophysics Laboratory, MGL; James Cook University; *https://www.marinegeophysics.com.au/current-meter*) instruments that measure velocities (u, v) and temperature parameters (Fig. 4.C). They work via a tilt sensor that uses

an accelerometer and a magnetometer to deduce current intensity and direction without considering about the orientation of the device. Within campaigns, Marotte HS instruments were used to obtain near-bottom current observations and were deployed in depths between 2 and 5 m in order to characterize fluxes generated by incident swells over the barrier reef or

circulation in shallow areas inside lagoons. By default, the instruments record at 0.5 Hz, this was subsequently averaged to provide a time series of near-bottom currents at 1 second frequency.

All ADCPs used within the project were from Teledyne RD Instruments Inc. (TRD-I), and were used to record a time series of velocity profiles over the water column as well as pressure and bottom temperature variations (Fig. 4.B). During Leg5 of the SPHYNX campaign, four Workhorse Sentinel 300 KHz (*http://www.teledynemarine.com/workhorse-sentinel-adcp?ProductLineID=12*) were deployed in the passages and on the forereef of Hienghène lagoon at depths ranging from 29 m to 32 m. Typical sampling rates were set to burst every 10 min, with a cell size of 0.3 m and 45 pings per ensemble. A

Sentinel V20 and a Sentinel V50 (1000 and 500 KHz respectively; http://www.teledynemarine.com/sentinel-v-adcp?ProductLineID=12) were deployed during the ELADE, CADHYAK and NEMO campaigns, targeting mainly areas of high expected currents (i.e., passes). These were deployed between 13 and 30 m depending on sites and aims. For the Sentinel V50, sampling rates were set to burst every 20 minutes with cells size equal to 0.5 m and 180 pings per ensemble, while the Sentinel V20, was set to burst at 10 minutes intervals with a 1 m cells size and 70 pings per ensemble.

To observe Lagrangian surface currents, reef drifters constructed by Pacific Gyre, Inc., (*https://www.pacificgyre.com/reef-drifter.aspx*) were deployed during each campaign (Fig. 4.A). Depending on their trajectories and the distance from coast, buoys were retrieved between 12 and 72 hours. These drifters are drogueless devices designed for shallow water use and are tough holed spheres with a GPS transponder that remain afloat at the surface. The reef drifters are set to transmit GPS position and surface temperature at 10 minutes intervals.

**4.3 Hydrological patterns: CTD profilers**

Two versions of Sea-Bird SeaCAT profilers (SBE 19plus and SBE 19plus V2; https://www.seabird.com/sbe-19plus-v2-seacat-profiler-ctd/product?id=60761421596) were used to measure the vertical and cross-shore patterns in temperature and salinity. Two auxiliary sensors were also equipped to our CTD profilers, namely, the WET Labs ECO-FLNTU and the Biospherical/Licor instrument which measure Fluorescence and Turbidity, and PAR respectively. The SBE19plus (used for

the SPHYNX and NOUMEA campaigns; Fig. 4.F) and the SBE19plus V2 (used for the CADHYAK and NEMO campaigns) both set to acquire data at 4 Hz. All water column (surface to bottom) profiles were done manually from coastal boats. For avoiding influence from the drift of the boat, only descent data were kept for final processing.

## 5    Processing and quality control

### 5.1 Overall strategy of processing

Data processing and quality control were conducted in a standardized manner, to ensure that observations from each campaign acts congruent data set. Maintenance and recalibration of instruments were conducted at recommended intervals by or in accordance with the manufacturers, to ensure reliability and quality of values observed. For entirety of the data set, a specific

nomenclature (of raw or processed files) was implemented such that filename contain the most important information about deployments: *Campaign-name_Station_Instrument-Model_Serial-number_Depth_Leg-Number.file-type.* In summary 380 processing and quality control of data consisted of applying the following steps:

1. Raw files were downloaded from instruments using the supplied manufacturer's software, renamed and saved on a secure network drive.

2. All data were carefully viewed using the supplied manufacturer's software to check for bad data (e.g., out of water, or outlier spikes) and to find the exact date and time of first and last good observations.

3. Data were then exported in the most convenient formats (depending on the software) to be read with Python 3. These formats were mostly ASCII or Matlab files.

4. Python 3 was used to process (when required) and convert data into NetCDF files. Metadata was also attached to each dataset during conversion to NetCDF, and includes information such as: beginning and end dates, the referring time, positions, depth, instrument model and serial number, project information, publisher information, contacts, processing 390 done, etc. During this step, data was also plotted and examined to ensure the quality of obtained time series.

5. Converted NetCDF data were subsequently plotted using Ferret to perform a last visual check, and metadata was also displayed and checked using command line NetCDF tools.

## 5.2 Specific processing steps

For each instrument type specific processing steps were included. For SBE56, raw data were averaged to provide a final 395 NetCDF time-series at 1 min. JFE data were averaged over the 10 samples from each burst, so that the final period is 10 or 15 minutes. For RBR pressure data, a barometric correction was applied with a constant atmospheric pressure (101325 bar). Because pressure sensors have all been deployed by scuba diving (sometimes in the vicinity of breaking waves, e.g., forereef) no precise vertical referencing by DGPS (Differential Global Positioning System) has been achieved. Furthermore, data have not been corrected from the long-term drifting of the sensor and, from drift due to loggers shift between legs. For waves 400 parameters, bursts were filtered using the Fourier transform to obtain a pressure spectrum (between 3 and 25 s period) then the linear wave theory is applied. This method uses a homogenous cut-off frequency of 0.33 to filter high frequencies spectrum (method used in Aucan et al., 2017). Furthermore, none of tail diagnostic have been applied on the higher frequency range but methods are existing (e.g., JONSWAP spectrum tail or TMA spectrum tail) (Karimpour and Chen, 2017). The processing method create two output files; one file containing temperature & level dynamics at 1 min resolution and the second at 1 hour 405 resolution offering computed wave parameters. ADCP data were filtered to avoid contaminated surface cells within the NetCDF data; final temporal resolution is thus the nominal resolution of the bursts. Marotte inclinometers data were averaged to 1 min sampling intervals to smooth out high frequency fluctuations in velocity, and orientation convention was changed to be congruent with oceanographic current convention. No specific treatment was applied to Pacific Gyre Reef Drifters, and the nominal frequency of 10 min was kept for the final data; drifter speed was also derived directly from the positions and included

in the final data. For CTD casts, classic processing steps using SeatermV2 (SEA-BIRD Electronics Inc.) software was applied and comprised of: data conversion into cnv, low-pass filtering, align-CTD, derive, and a bin averaging at 0.5 m, respectively. The final data set consists of 5 individual data series, one for each sampled lagoon (see Section 7).

## 6 Overview of observations

Figure 5 highlights a synopsis of scientific objectives or processes evoked in section 3.1. Figure 5.A presents thermal variability
over external reefs slopes for three campaigns (SPHYNX, CADHYAK, NEMO). Although this plot shows different years, it enables visualization of the inter-site variability in terms of oceanic processes. During SPHYNX summer season, for example, cyclic pulses of fresh water that struck station C02 (SBE56 – depth 26 m) inducing thermal variability. A thorough analysis (not shown here) showed that drops in temperature where around the M2 tidal waves frequency and that it was observable only in unusual wind conditions (either very few winds or North-West winds). Those facts lead us to consider this area as an
arrival point for internal waves, which might not appear in data when trade winds are present due to inducing favorable conditions for downwelling. During the NEMO campaign, from 19[th] to 26[th] of January 2021, a significant decrease in temperature happened at station R03 (RBRduet T.D – depth 9.7 m), reaching almost 5 degrees which was probably the signature of an intense upwelling event on the external slope of Moindou lagoon. Finally, the station R03 (RBR Duet – depth 11.7 m) displays a temperature decrease during the passing of TC "Uesi" that struck during CADHYAK campaign. TC "Ola"
and TD "Lucas" did not seem to have significantly altered the oceanic temperature dynamics during SPHYNX and NEMO respectively.

Figure 5.B displays the salinity structure of the Hienghène-Touho lagoon on cross-shore section T in December 2014, June 2015 and March 2016. T07 to T13 stations are more in the vicinity of the large passages on the North-West side of Hienghène-
Touho lagoon and thus are more influenced by oceanic waters. The two first plots reveal the difference in haline structuration between seasons with a gradient of approx. 0.2 PSU along the coast. At the end of the dry season (December 2014), coastal stations were saltier than the stations in or closer to open ocean, while it is exactly the opposite at the end of the wet season (June 2015). The March 2016 radial offers a view of the spatial and vertical extension of the river plumes during this campaign.

Fig. 5.C presents the trajectories of the buoys realized during SEARSE and SAR experiments in Nouméa lagoon. These drifter trajectories highlight the strong dependency of surface circulation to wind forcings in New Caledonian lagoons (e.g., February 2020 versus April 2020 SEARSE releases, February 2020 SEARSE release versus November 2019 green trajectory of SAR release). The cyan trajectory (SAR experiment) shows an interesting feature of the South-Western lagoon of New Caledonia which is consistent with an anti-cyclonic gyre in this portion of the lagoon (Ouillon et al., 2010).

Finally, Fig 5. D is centered on the "Oma" cyclone that hit New Caledonia early 2019. On the external slope of Poé lagoon (Fig 5.D.1), this event generated huge incident waves reaching a significant wave height of ~ 6 m at the climax of the perturbation (RBRduet T.D – O2 station – depth 10.8 m). These strong sea-states also induced an intense surge of 0.6 m inside the lagoon (RBRduo T.D – L13 station – depth 2.2 m). Inclinometers located at the back of the reef crest (Marotte HS – Station L06 – depth 2 m) recorded fluxes twice more intense as the normal conditions, reaching 0.5 m s$^{-1}$. Finally, ADCP Sentinel V50 (Station P01 – depth 30 m) also showed a significant increase over the whole water column during this event and the Shark fault circulation was not anymore linked to tidal cycle but only outgoing and flushing waters toward oceanic side (not shown here).

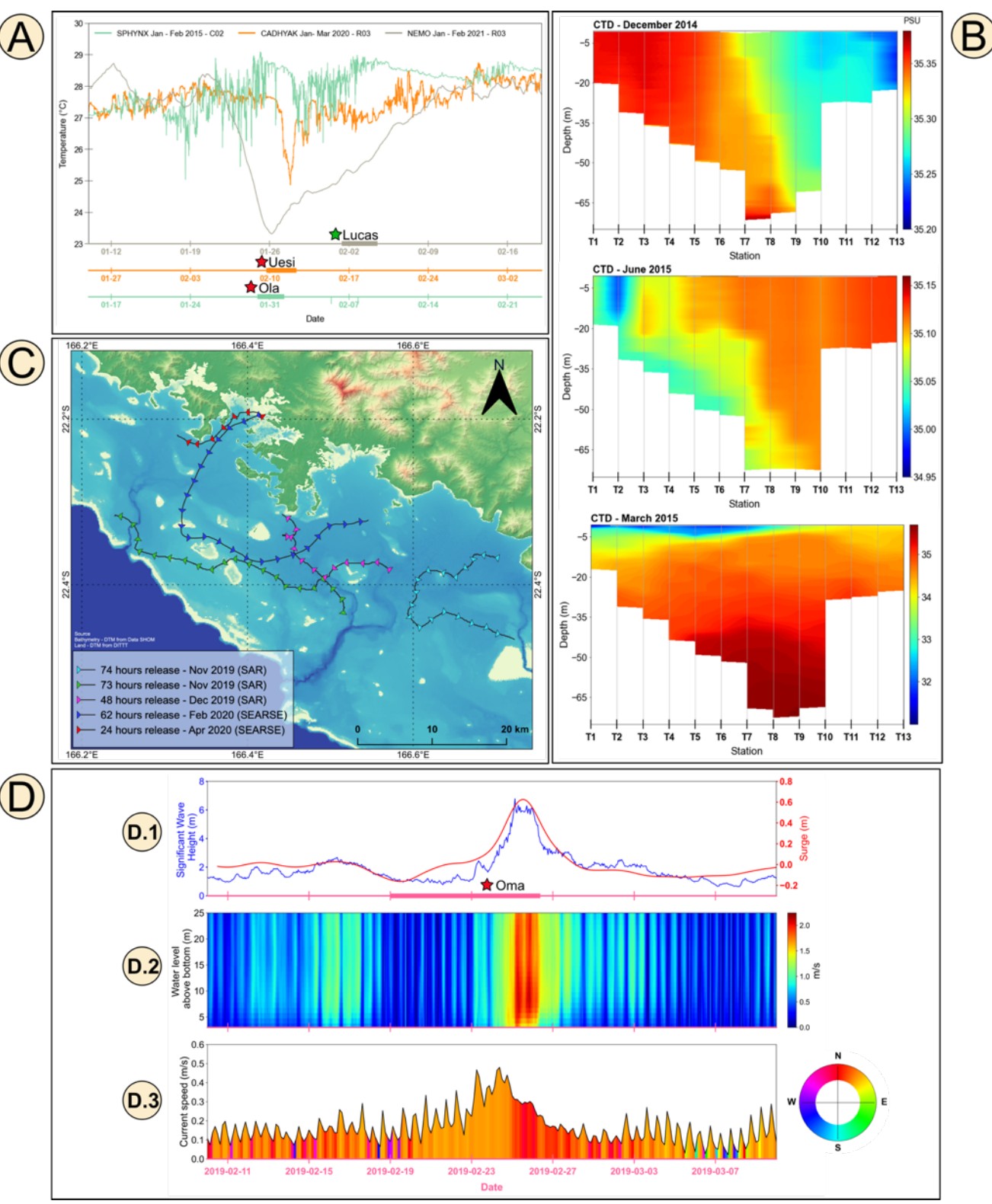

**Figure 5: Overview of some physical and hydrodynamical processes occurred over PRESENCE project: (A) Presentation of temporal variation of temperature during SPHYNX, CADHYAK and NEMO campaigns. (B) Cross-shore expansion and vertical stratification of salinity in Hienghène-Touho lagoon (SPHYNX campaign). (C) Lagrangian circulation recorded with Reef Drifters in Nouméa area (SEARSE and SAR experiments). (D) Cross-reef and passage circulation in Poé lagoon (ELADE campaign): (D.1) Significant Wave Height (Hsig) at O2 station and surge at L13 station; (D.2) Current speed over water column in the shark fault**
**(P01 station); (D.3) Current speed measured inside lagoon at L06 station.**

## 7 Data availability

Data sets from all 5 campaigns presented herein are freely available on SEANOE (https://www.seanoe.org/) in dedicated repositories and exist in NetCDF format. Data from the SAR and SEARSE experiments are available on the Sextant repository (https://sextant.ifremer.fr) in shapefile format. DOI's for individual data sets are provided in Table 1. Survey reports are

460 available in French on the Archimer (https://archimer.ifremer.fr/) repository and present a generic overview of the sampling region, the meteorological and oceanic conditions as well as raw representation of the observations acquired. These data sets are available for and are already being used for investigations into the coastal dynamics conjointly with hydrodynamics models around Grande Terre, and publications are in preparation.

**Table 1: DOI links and references of data sets and associated field reports presented in this paper**

| Campaign | Data set | | Archimer | |
|---|---|---|---|---|
| | DOI | Reference | DOI | Reference |
| **SPHYNX** | https://doi.org/10.17882/54005 | Le Gendre et al., (2018) | https://doi.org/10.13155/56378 | Le Gendre et al., (2018) |
| **NOUMÉA** | https://doi.org/10.17882/53974 | Desclaux et al., (2017) | https://doi.org/10.13155/58315 | Desclaux et al., (2018) |
| **ELADE** | https://doi.org/10.17882/76334 | Le Gendre et al., (2020) | / | / |
| **CADHYAK** | https://doi.org/10.17882/79616 | Bruyère et al., (2021) | http://dx.doi.org/10.13155/80615 | Bruyère et al., (2021) |
| **NEMO** | https://doi.org/10.17882/81063 | Bruyère et al., (2021) | https://doi.org/10.13155/87118 | Bruyère et al., (2022) |
| **SEARSE** | https://doi.org/10.12770/fa99ffe5-83e5-483c-8065-90a79981140a | Soulard, (2019) | / | / |
| | https://doi.org/10.12770/dad19639- | Soulard, (2020) | | |

| | c901-4edb-85cd-1fd546aa4cdb | | | |
|---|---|---|---|---|
| **SAR** | http://dx.doi.org/10.12770/96e4f2ef-e809-4005-b5df-529adc4e3306 | Soulard et al., (2021) | / | / |

## 8 Conclusion

The PRESENCE data sets presented in this paper make available a remarkable and unique opportunity to investigate lagoon hydrodynamics by being freely accessible and composed of observations from five different lagoons around Grande Terre during cyclone seasons. Eight major atmospheric events occurred during our observation period (2014-2021) and sampling
strategies employed enabled us to capture their signatures in various lagoons and their adjacent forereef (e.g., incident swells). It represents a considerable contribution towards knowledge on lagoon dynamics around New Caledonia, which will allow easier investigation and assist with assessment of numerical model experiments or satellite derived parameters in the future.

The spatial extent of all sampling strategies offers a synoptic view of distinct lagoon functioning and dynamics, and complements the existing Reeftemps network. Furthermore, even though the five lagoons were not sampled simultaneously,
this data set presents an important milestone about the knowledge on processes that take place during cyclonic events.

Various interesting features were also highlighted by the availability of this long-term observational strategy, and include thermal dynamics identifying internal waves and upwelling, distinct surface circulation (such as, SW lagoon gyre, wind dependency drift), land-lagoon continuum variability brought about by river plumes dynamics or through hydrological structuration and quantification of ocean-lagoon exchanges over reef crests or through passes. Finally, the data set presented
through the PRESENCE project will provide a unique opportunity to gain deeper understanding of coastal vulnerability or impacts of Marine Heat Waves on coastal New Caledonian lagoons and helps to fill a Pacific Island wide data and knowledge gap for coastal hydrodynamics.

### Author contributions:

RLG, TL, HLM and BS set up the PRESENCE project and raised funds. RLG and BS designed and conducted the experiments
as principal investigators. All co-authors were implied in some of the field experiments. OB and RLG organized, processed, checked and archived the data sets. OB and RLG prepared the paper and designed the figures, with contributions from all co-authors.

**Competing interest:**

The authors declare that there are no competing interests associated with this study.

**Acknowledgements:**

The authors acknowledge the New Caledonia institutions for their financial support. We thank the following diving centers and taxi boat operators for their field knowledge and services at sea, Babou Océan at Hienghène, Rêve bleu diving at Koumac and Romain Laigle at Bourail as well as all divers involved: Gilbert Sarrailh, Eric Folcher, Bertrand Bourgeois, Armelle Renaud. We also deeply thank David Varillon and Céline Bachelier from US IMAGO (IRD) for sharing their experiences and 495    materials. Further acknowledgment is extended to Jérôme Aucan from UMR ENTROPIE (IRD) for sharing his Python script using linear theory reconstruction of wave parameters. A particular thank you is extended to Shilpa Lal and Jasha Dehm for their extensive proofreading and review of the language used.

**Financial support:**

This research has been supported entirely by the New Caledonian Institutions (Government, North and South Provinces). 500    Additional instruments used were financed by the Consortium for the research, higher education and innovation in New Caledonia (CRESICA).

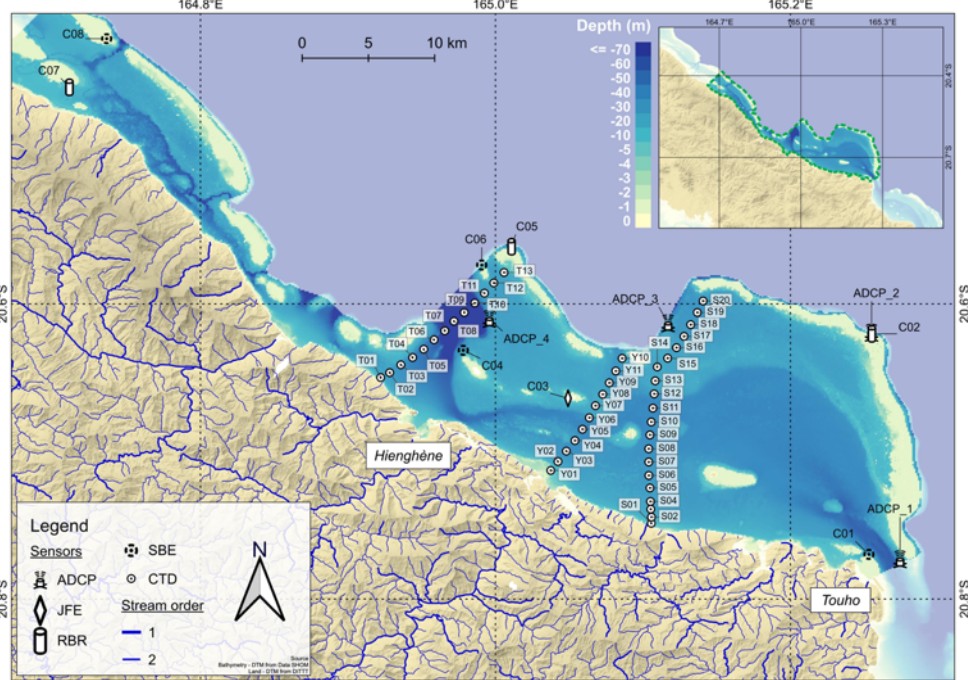

**Fig. A1: Observational strategy deployed during SPHYNX field experiment. ADCP: Acoustic Doppler Current profiler; JFE: INFINITY-CTW; RBR: RBRduo T.D or RBRduet T.D; CTD: Conductivity – Temperature – Depth ; SBE: SBE 56.**

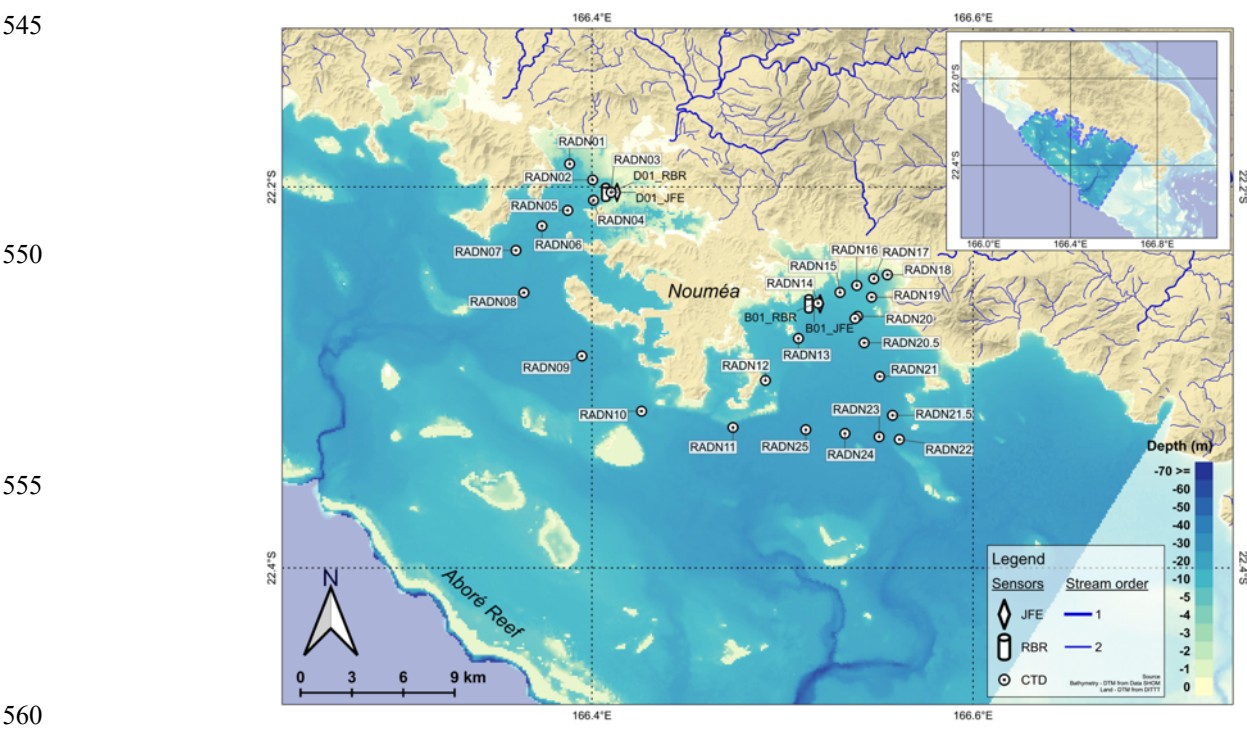

**Fig. B1: Observational strategy deployed during NOUMEA field experiment. JFE: INFINITY-CTW; RBR: RBRduo T.D or RBRduet T.D; CTD: Conductivity – Temperature – Depth.**

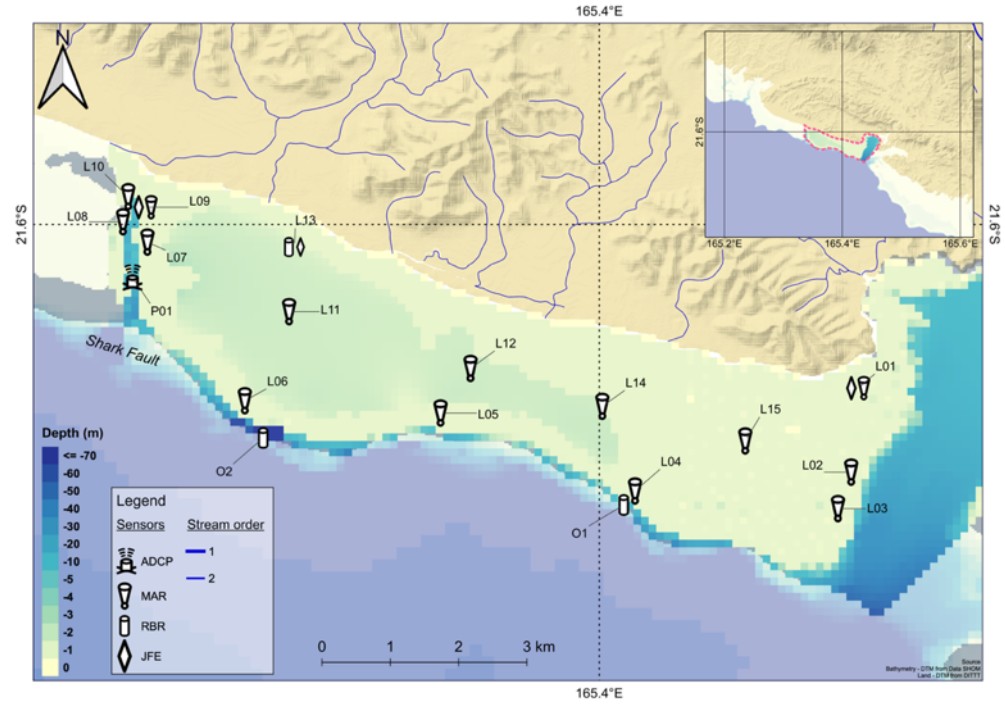

**Fig. C1: Observational strategy deployed during ELADE field experiment. ADCP: Acoustic Doppler Current profiler; JFE: INFINITY-CTW; MAR: Marotte HS; RBR: RBRduo T.D or RBRduet T.D.**

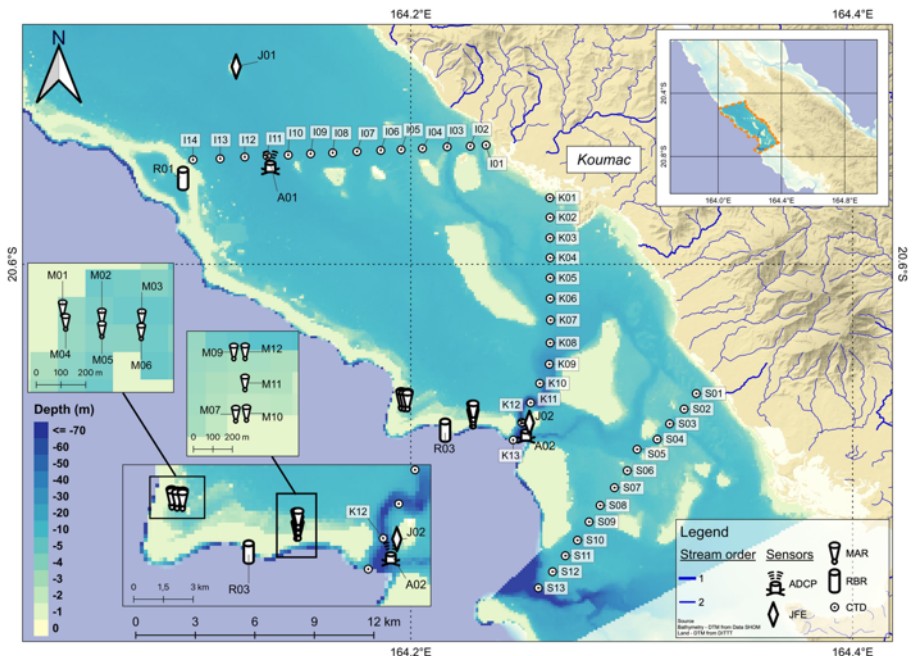

**Fig. D1: Observational strategy deployed during CADHYAK field experiment. ADCP: Acoustic Doppler Current profiler; JFE: INFINITY-CTW; MAR: Marotte HS; RBR: RBRduo T.D or RBRduet T.D; CTD: Conductivity – Temperature – Depth.**

**Table. A1: Moorings and loggers information deployed during PRESENCE surveys.**

| Station | Instrument | Raw parameters | Longitude (W) | Latitude (S) | Date Start | Date End | Freq | Processed parameters | Legs |
|---|---|---|---|---|---|---|---|---|---|
| **SPHYNX** | | | | | | | | | |
| C01 | SBE56 | Temperature | 165.2532000 | 20.7697333 | 02/12/2014 | 07/03/2016 | 30s | Temperature | 1,2,3,4,5 |
| C02 | RBR Duo T.D | Temperature - Pressure | 165.2551833 | 20.6202666 | 02/12/2014 | 08/03/2016 | 1 Hz | Temperature - Wave Height & Period - Water Level | 1,2,3,4,5 |
| C03 | JFE ACTW-USB | Temperature - Conductivity | 165.0492500 | 20.6639666 | 04/12/2014 | 07/03/2016 | 10 min | Temperature - Salinity | 1,2,3,4,5 |
| C04 | SBE56 | Temperature | 164.9782000 | 20.6314166 | 03/12/2014 | 08/03/2016 | 30s | Temperature | 1,2,3,4,5 |
| C05 | RBR Duo T.D | Temperature - Pressure | 165.0109333 | 20.5612333 | 04/12/2014 | 08/03/2016 | 1 Hz | Temperature - Wave Height & Period - Water Level | 1,2,3,4,5 |
| C06 | SBE56 | Temperature | 164.9907000 | 20.573633 | 04/12/2014 | 09/03/2016 | 30s | Temperature | 1,2,3,4,5 |
| C07 | RBR Duo T.D | Temperature - Pressure | 164.7111500 | 20.4531833 | 05/12/2014 | 05/03/2015 | 1 Hz | Temperature - Wave Height & Period - Water Level | 1 |
| C08 | SBE56 | Temperature | 164.7364167 | 20.4202333 | 06/12/2014 | 02/09/2015 | 30s | Temperature | 1,2,3 |
| ADCP_4 | Workhorse ADCP Sentinel 300 kHz | Current - Temperature - Pressure | 164.9958867 | 20.6101766 | 05/01/2016 | 09/03/2016 | 10 min | Temperature - Current Speed & Direction - Water Level | 5 |
| ADCP_1 | Workhorse ADCP Sentinel 300 kHz | Current - Temperature - Pressure | 165.2743333 | 20.7730833 | 04/01/2016 | 07/03/2016 | 10 min | Temperature - Current Speed & Direction - Water Level | 5 |
| ADCP_2 | Workhorse ADCP Sentinel 300 kHz | Current - Temperature - Pressure | 165.2550000 | 20.6195333 | 04/01/2016 | 08/03/2016 | 10 min | Temperature - Current Speed & Direction - Water Level | 5 |
| ADCP_3 | Workhorse ADCP Sentinel 300 kHz | Current - Temperature - Pressure | 165.1170833 | 20.6131833 | 05/01/2016 | 08/03/2016 | 10 min | Temperature - Current Speed & Direction - Water Level | 5 |

| Station | Instrument | Raw parameters | Longitude (W) | Latitude (S) | Date Start | Date End | Freq | Processed parameters | Legs |
|---|---|---|---|---|---|---|---|---|---|
| **NOUMEA** | | | | | | | | | |
| B01_JFE | JFE ACTW-USB | Temperature - Conductivity | 166.51665 | 22.2613500 | 02/12/2016 | 25/04/2017 | 15 min | Temperature - Salinity | 1 |
| B01_RBR | RBR Duo T.D | Temperature - Pressure | 166.51665 | 22.2613500 | 02/12/2016 | 29/03/2017 | 1 Hz | Temperature - Wave Height & Period - Water Level | 1 |
| D01_JFE | JFE ACTW-USB | Temperature - Conductivity | 166.410112 | 22.202842 | 02/12/2016 | 25/04/2017 | 15 min | Temperature - Salinity | 1 |
| D01_RBR | RBR Duo T.D | Temperature - Pressure | 166.410112 | 22.202842 | 02/12/2016 | 29/03/2017 | 1 Hz | Temperature - Wave Height & Period - Water Level | 1 |

| | | | | | | | | | |
|---|---|---|---|---|---|---|---|---|---|
| **ELADE** | | | | | | | | | |
| L01 | JFE ACTW-USB | Temperature - Conductivity | 165.43405 | 21.62155 | 18/03/2019 | 01/09/2019 | 10 min | Temperature - Salinity | 1,2 |
| | Marotte HS | Current - Temperature | 165.43405 | 21.62155 | 04/02/2019 | 01/09/2019 | 1 min | Temperature - Current Speed & Direction | 1,2 |
| L02 | Marotte HS | Current - Temperature | 165.43314 | 21.63261 | 04/02/2019 | 01/09/2019 | 1 min | Temperature - Current Speed & Direction | 1,2 |
| L03 | Marotte HS | Current - Temperature | 165.4314 | 21.63741 | 05/02/2019 | 01/09/2019 | 10s | Temperature - Current Speed & Direction | 1,2 |
| L04 | Marotte HS | Current - Temperature | 165.40472 | 21.63509 | 06/02/2019 | 01/09/2019 | 10s | Temperature - Current Speed & Direction | 1,2 |
| L05 | Marotte HS | Current - Temperature | 165.37915 | 21.62483 | 06/02/2019 | 01/09/2019 | 10s | Temperature - Current Speed & Direction | 1,2 |
| L06 | Marotte HS | Current - Temperature | 165.35342 | 21.62321 | 06/02/2019 | 01/09/2019 | 10s | Temperature - Current Speed & Direction | 1,2 |
| L07 | Marotte HS | Current - Temperature | 165.34061 | 21.60232 | 06/02/2019 | 01/09/2019 | 10s | Temperature - Current Speed & Direction | 1,2 |
| L08 | Marotte HS | Current - Temperature | 165.3374 | 21.59965 | 06/02/2019 | 07/05/2019 | 10s | Temperature - Current Speed & Direction | 1 |
| L09 | Marotte HS | Current - Temperature | 165.34035 | 21.59773 | 06/02/2019 | 01/09/2019 | 10s | Temperature - Current Speed & Direction | 1,2 |
| | JFE ACTW-USB | Temperature - Conductivity | 165.34035 | 21.59773 | 18/03/2019 | 07/05/2019 | 10 min | Temperature - Salinite | 1 |
| L10 | Marotte HS | Current - Temperature | 165.33809 | 21.59634 | 06/02/2019 | 07/05/2019 | 10s | Temperature - Current Speed & Direction | 1 |
| L11 | Marotte HS | Current - Temperature | 165.35924 | 21.61148 | 06/02/2019 | 01/09/2019 | 10s | Temperature - Current Speed & Direction | 1,2 |
| L12 | Marotte HS | Current - Temperature | 165.38309 | 21.61896 | 05/02/2019 | 01/09/2019 | 10s | Temperature - Current Speed & Direction | 1,2 |
| L13 | RBR Duo T.D | Temperature - Pressure | 165.35994 | 21.60297 | 08/02/2019 | 05/05/2019 | 1 Hz | Temperature - Wave Height & Period - Water Level | 1 |
| L14 | Marotte HS | Current - Temperature | 165.40042 | 21.623945 | 01/07/2019 | 01/09/2019 | 10s | Temperature - Current Speed & Direction | 2 |
| L15 | Marotte HS | Current - Temperature | 165.41921 | 21.6285 | 01/07/2019 | 01/09/2019 | 10s | Temperature - Current Speed & Direction | 2 |
| O1 | RBR Duet T.D | Temperature - Pressure | 165.40324 | 21.63693 | 05/02/2019 | 07/05/2019 | 1Hz | Temperature - Wave Height & Period - Water Level | 1 |
| O2 | RBR Duet T.D | Temperature - Pressure | 165.35583 | 21.6281 | 05/02/2019 | 07/05/2019 | 1Hz | Temperature - Wave Height & Period - Water Level | 1 |
| P01 | ADCP Sentinel-V20 | Current - Temperature - Pressure | 165.33865 | 21.60699 | 05/02/2019 | 08/05/2019 | 10 min | Temperature - Current Speed & Direction - Water Level | 1 |
| B21 | RBR Duet T.D | Temperature - Pressure | 165.35994 | 21.60297 | 01/07/2019 | 01/09/2019 | 10s | Temperature - Wave Height & Period - Water Level | 2 |
| | JFE ACTW-USB | Temperature - Conductivity | 165.35994 | 21.60297 | 01/07/2019 | 01/09/2019 | 10 min | Temperature - Salinity | 2 |

| | | | | | | | | | | |
|---|---|---|---|---|---|---|---|---|---|---|
| | **CADHYAK** | | | | | | | | | |
| A01 | ADCP Sentinel-V20 | Current - Temperature - Pressure | 164.136466 | 20.554121 | 10/12/2019 | 06/05/2020 | 20 min | Temperature - Current Speed & Direction - Water Level | 1 |
| A02 | ADCP Sentinel-V50 | Current - Temperature - Pressure | 164.252161 | 20.675672 | 11/12/2019 | 18/05/2020 | 10 min | Temperature - Current Speed & Direction - Water Level | 1 |
| J01 | JFE ACTW-USB | Temperature - Conductivity | 164.120939 | 20.510531 | 12/12/2019 | 26/05/2020 | 15 min | Temperature - Salinity | 1 |
| J02 | JFE ACTW-USB | Temperature - Conductivity | 164.253714 | 20.671781 | 12/12/2019 | 27/05/2020 | 15 min | Temperature - Salinity | 1 |
| M01 | Marotte HS | Current - Temperature | 164.19567 | 20.660921 | 10/12/2019 | 28/05/2020 | 1 Hz | Temperature - Current Speed & Direction | 1 |
| M02 | Marotte HS | Current - Temperature | 164.197096 | 20.66123 | 10/12/2019 | 28/05/2020 | 1 Hz | Temperature - Current Speed & Direction | 1 |
| M03 | Marotte HS | Current - Temperature | 164.198547 | 20.661251 | 10/12/2019 | 01/05/2020 | 1 Hz | Temperature - Current Speed & Direction | 1 |
| M04 | Marotte HS | Current - Temperature | 164.195782 | 20.661425 | 10/12/2019 | 19/05/2020 | 1 Hz | Temperature - Current Speed & Direction | 1 |
| M05 | Marotte HS | Current - Temperature | 164.197097 | 20.661697 | 10/12/2019 | 28/05/2020 | 1 Hz | Temperature - Current Speed & Direction | 1 |
| M06 | Marotte HS | Current - Temperature | 164.198529 | 20.661771 | 10/12/2019 | 28/05/2020 | 1 Hz | Temperature - Current Speed & Direction | 1 |
| M07 | Marotte HS | Current - Temperature | 164.22789 | 20.669449 | 10/12/2019 | 28/05/2020 | 1 Hz | Temperature - Current Speed & Direction | 1 |
| M09 | Marotte HS | Current - Temperature | 164.227801 | 20.666582 | 10/12/2019 | 16/05/2020 | 1 Hz | Temperature - Current Speed & Direction | 1 |
| M10 | Marotte HS | Current - Temperature | 164.228385 | 20.669414 | 10/12/2019 | 28/05/2020 | 1 Hz | Temperature - Current Speed & Direction | 1 |
| M11 | Marotte HS | Current - Temperature | 164.228316 | 20.668009 | 10/12/2019 | 28/05/2020 | 1 Hz | Temperature - Current Speed & Direction | 1 |
| M12 | Marotte HS | Current - Temperature | 164.228321 | 20.666574 | 10/12/2019 | 28/05/2020 | 1 Hz | Temperature - Current Speed & Direction | 1 |
| R01 | RBR Duo T.D | Temperature - Pressure | 164.096938 | 20.561161 | 12/12/2019 | 28/04/2020 | 1 Hz | Temperature - Wave Height & Period - Water Level | 1 |
| R03 | RBR Duet T.D | Temperature - Pressure | 164.21555 | 20.675113 | 12/12/2019 | 06/04/2020 | 1 Hz | Temperature - Wave Height & Period - Water Level | 1 |

| | | NEMO | | | | | | | |
|---|---|---|---|---|---|---|---|---|---|
| A01 | ADCP Sentinel-V20 | Current - Temperature - Pressure | 165.466447 | 21.652592 | 07/09/2020 | 02/12/2020 | 10 min | Temperature - Current Speed & Direction - Water Level | 1 |
| A02 | ADCP Sentinel-V20 | Current - Temperature - Pressure | 165.641488 | 21.767753 | 07/09/2020 | 01/12/2020 | 10 min | Temperature - Current Speed & Direction - Water Level | 1 |
| A03 | ADCP Sentinel-V20 | Current - Temperature - Pressure | 165.551992 | 21.739817 | 04/12/2020 | 28/04/2021 | 10 min | Temperature - Current Speed & Direction - Water Level | 2 |
| A04 | ADCP Sentinel-V20 | Current - Temperature - Pressure | 165.462596 | 21.688893 | 04/12/2020 | 29/04/2021 | 10 min | Temperature - Current Speed & Direction - Water Level | 2 |
| J01 | JFE ACTW-USB | Temperature - Conductivity | 165.463297 | 21.653583 | 08/09/2020 | 19/04/2021 | 10 min | Temperature - Salinity | 1,2 |
| J02 | JFE ACTW-USB | Temperature - Conductivity | 165.487136 | 21.681908 | 08/09/2020 | 25/04/2021 | 10 min | Temperature - Salinity | 1,2 |
| J03 | JFE ACTW-USB | Temperature - Conductivity | 165.604377 | 21.749468 | 08/09/2020 | 28/04/2021 | 10 min | Temperature - Salinity | 1,2 |
| M01 | Marotte HS | Current - Temperature | 165.463281 | 21.653552 | 06/09/2020 | 29/04/2021 | 1 Hz | Temperature - Current Speed & Direction | 1,2 |
| M02 | Marotte HS | Current - Temperature | 165.4601 | 21.655466 | 06/09/2020 | 21/03/2021 | 1 Hz | Temperature - Current Speed & Direction | 1,2 |
| M03 | Marotte HS | Current - Temperature | 165.454915 | 21.659515 | 06/09/2020 | 24/01/2021 | 1 Hz | Temperature - Current Speed & Direction | 1,2 |
| M04 | Marotte HS | Current - Temperature | 165.467313 | 21.690576 | 06/09/2020 | 29/04/2021 | 1 Hz | Temperature - Current Speed & Direction | 1,2 |
| M05 | Marotte HS | Current - Temperature | 165.480559 | 21.704749 | 06/09/2020 | 15/01/2021 | 1 Hz | Temperature - Current Speed & Direction | 1,2 |
| M06 | Marotte HS | Current - Temperature | 165.506993 | 21.724816 | 06/09/2020 | 28/04/2021 | 1 Hz | Temperature - Current Speed & Direction | 1,2 |
| M07 | Marotte HS | Current - Temperature | 165.511434 | 21.74525 | 06/09/2020 | 31/01/2021 | 1 Hz | Temperature - Current Speed & Direction | 1,2 |
| M08 | Marotte HS | Current - Temperature | 165.555531 | 21.767376 | 06/09/2020 | 05/03/2021 | 1 Hz | Temperature - Current Speed & Direction | 1,2 |
| M09 | Marotte HS | Current - Temperature | 165.604377 | 21.749468 | 05/09/2020 | 20/02/2021 | 1 Hz | Temperature - Current Speed & Direction | 1,2 |
| M10 | Marotte HS | Current - Temperature | 165.596128 | 21.772948 | 07/09/2020 | 10/01/2021 | 1 Hz | Temperature - Current Speed & Direction | 1,2 |
| M11 | Marotte HS | Current - Temperature | 165.61953 | 21.785012 | 05/09/2020 | 28/04/2021 | 1 Hz | Temperature - Current Speed & Direction | 1,2 |
| R01 | RBR Duet T.D | Temperature - Pressure | 165.477092 | 21.704961 | 08/09/2020 | 15/06/2021 | 1 Hz | Temperature - Wave Height & Period - Water Level | 1,2 |
| R02 | RBR DuO T.D | Temperature - Pressure | 165.554976 | 21.739488 | 08/09/2020 | 28/04/2021 | 1 Hz | Temperature - Wave Height & Period - Water Level | 1,2 |
| R03 | RBR Duet T.D | Temperature - Pressure | 165.555728 | 21.77153 | 08/09/2020 | 11/06/2021 | 1 Hz | Temperature - Wave Height & Period - Water Level | 1,2 |

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
