# Peer review of "Hydrodynamic and hydrological processes within a variety of coral reef lagoons: Field observations during 6 cyclonic seasons in New Caledonia"

_Earth System Science Data, 2022_

## Referee Comment (RC1)

**Review of «Hydrodynamic and hydrological processes within a variety of coral reef lagoons: Field observations during 6 cyclonic seasons in New Caledonia» by Bruyère et al.**

Submitted to Earth System Science Data

Reviewed by Damien Sous

This manuscript reports on an extensive in-situ monitoring of lagoon hydrodynamics in New Caledonia, France. The research project is of great interest, well placed in the global effort to document reef-lagoon systems in growing threat context. It appears to be conducted with rigour, in particular considering the remote and harsh access of studied sites. The bibliography is extensive and well replaced in the present context. The manuscript is globally well illustrated. I think this paper may be a reference document for future studies about NC reef-lagoon systems. However, it is not suitable for publication in the present form. My main concerns are (see below) : (i) careful proofreading, (ii) lack of synthesized information about instruments and processed parameters, (iii) absence of method description for several parameters of primary importance (waves, levels).

Main remarks

- I do suggest an extensive proofreading by a native english speaker. I will not go into detailed corrections, but many sentences (while not grammatically false) are heavy and/or blurry.

- l.170 : which plumes ?

- «ocean reef slope » → « forereef », everywhere

- Top and bottom plots in Figure 1 should be separated, with separated captions

- A general instrumentation table is missing, recalling the main informations (type, dates, position, parameters, measurement timing) of all deployments in a given place.

- Similarly, there is a lack for a precise list (table) of processed parameters (sea level, wave height, temperature, etc) with related processing parameters.

- Marotte HS : how the data produced by drag-tilt bottom currentmeters can be interpreted in the presence of strong reef-induced friction and associated bottom boundary layer ? My understanding is the measured data will be ok for clear sandy area, but much less reliable in the presence of coral.

- Sea level : the precise measurement of sea-level is a tricky issue. This is apparently one of the processed parameter, but nothing is said about the sea level reconstruction : vertical positioning of the sensor, compensation of drift, etc

- Wave height : similarly to Sea level, there is a lack of detailed information about the reconstruction of wave features from the measured bottom pressure.

---

## Author Response (AR1)

**Response to Reviewers**

**Reviewer #1:**

This manuscript reports on an extensive in-situ monitoring of lagoon hydrodynamics in New Caledonia, France. The research project is of great interest, well placed in the global effort to document reef-lagoon systems in growing threat context. It appears to be conducted with rigor, in particular considering the remote and harsh access of studied sites. The bibliography is extensive and well replaced in the present context. The manuscript is globally well illustrated. I think this paper may be a reference document for future studies about NC reef-lagoon systems. However, it is not suitable for publication in the present form. My main concerns are (see below): (i) careful proofreading, (ii) lack of synthesized information about instruments and processed parameters, (iii) absence of method description for several parameters of primary importance (waves, levels).

Comment #1 of Reviewer #1 (in pdf file): I do suggest an extensive proofreading by a native English speaker. I will not go into detailed corrections, but many sentences (while not grammatically false) are heavy and/or blurry.

Answer to Comment #1 of Reviewer #1: *Manuscript has been extensively proofread by 2 natives E.S who enhanced the quality of English sentence (~50 % of the sentences rewritten/modified). We also added our two kinds English reviewers in the acknowledgement section.*

Comment #2 of Reviewer #1 (in pdf file): l.170 : which plumes ?

Answer to Comment #2 of Reviewer #1: *We thank the reviewer to have noticed this imprecision. We modified the sentence as follows: "Finally, the fate of rivers plumes (from the Dumbéa, Coulée, Pirogues rivers) and their consequences on the SW lagoon were studied through biogeochemical and sedimentological studies conducted by Pinazo et al., 2004; Ouillon et al., 2004; Drouzy et al., 2019" (L178 - L180) to precise the concerned watersheds.*

Comment #3 of Reviewer #1 (in pdf file): «ocean reef slope » « → forereef », everywhere

Answer to Comment #3 of Reviewer #1: *We followed the recommendation of Reviewer #1 and we modified each "ocean reef slope" to forereef word at four occurrences.*

Comment #4 of Reviewer #1 (in pdf file): Top and bottom plots in Figure 1 should be separated, with separated captions

Answer to Comment #4 of Reviewer #1: *We agree with this comment and we split Figure 1 into 2 separate figures. References to figures are now modified within 3.2 section that describe lagoons morphologies and sampling strategies.*

Comment #5 of Reviewer #1 (in pdf file): A general instrumentation table is missing, recalling the main informations (type, dates, position, parameters, measurement timing) of all deployments in a given place.

Answer to Comment #5 of Reviewer #1: *We agree with this comment of Reviewer #1 and we added a full table in appendices section giving main information about deployment. In lines 308 to 309 we added a sentence referring to the Table in supplementary (Table. A1)*

Comment #6 of Reviewer #1 (in pdf file): Similarly, there is a lack for a precise list (table) of processed parameters (sea level, wave height, temperature, etc) with related processing parameters.

Answer to Comment #6 of Reviewer #1: *Processed parameters now appear in Table A1 in the Appendices section (see answer of comment 5).*

Comment #7 of Reviewer #1 (in pdf file): Marotte HS: how the data produced by drag-tilt bottom currentmeters can be interpreted in the presence of strong reef-induced friction and associated bottom boundary layer? My understanding is the measured data will be ok for clear sandy area, but much less reliable in the presence of coral.

Answer to Comment #7 of Reviewer #1: *We thank Reviewer #1 for this comment, nevertheless Marotte HS have often been used in coral reef ecosystems (see following references: Faivre et al., 2020; Page et al., 2021; Blacka et al., 2019). These studies used Marotte HS current meter in coral reef habitats and highlighted the fact that drag-tilt current meter allow to investigate water flow inside coral environments and give and accurate temperature and velocity near the boundary contrary to standard ADCP's for example. The Marotte HS manufacturer also defends deployment in reef ecosystems to measure local current caused by reef structure, (see: https://www.marinegeophysics.com.au/current-meter/). Regardless, during our surveys majority of our loggers were moored on a sandy bed but this data logger can be deployed in areas that are topographically complex.*
*We chose to work primarily with Marotte HS current meters as they are affordable and easier to deployed than ADCPs, making widespread sampling more feasible, for further surveys we recommend the use of Aquadopp Profilers, which have now acquired and hence plan to perform more accurate data in such rugose coral environment.*

Comment #8 of Reviewer #1 (in pdf file): Sea level: the precise measurement of sea-level is a tricky issue. This is apparently one of the processed parameter, but nothing is said about the sea level reconstruction: vertical positioning of the sensor, compensation of drift, etc…

Answer to Comment #8 of Reviewer #1: *We thank reviewer #1 for this comment and based on this we added inside the manuscript information about the sea level parameter and reconstruction, as follows: "For RBR pressure data, a barometric correction was applied with a constant atmospheric pressure (101325 bar). Because pressure sensors have all been deployed by scuba diving (sometimes in the vicinity of breaking waves, e.g., forereef) no precise vertical referencing by DGPS (Differential Global Positioning System) has been achieved. Furthermore, data have not been corrected from the long-term drifting of the sensor and, from drift due to loggers shift between legs" (lines 396 to 399).*

*For consistency, we decided to do not remove the long-term drifting error in pressure time series, as most pressure gauges were started using delayed start time. Thus, loggers did not record any pressure data before immersion which is a prerequisite for long-term drift*

*compensation (e.g., Sous et al., 2020). For other processing strategies, raw data may be diffused on demand.*

Comment #9 of Reviewer #1 (in pdf file): Wave height: similarly, to Sea level, there is a lack of detailed information about the reconstruction of wave features from the measured bottom pressure.

Answer to Comment #9 of Reviewer #1: *We agree with Reviewer #1 for the comment 9 and to facilitate this we added in lines 399 to 403 detailed information of the method used for the linear wave theory. We chose to use the method explained in Aucan et al., 2017 which takes a constant cut-off frequency of 0.33 and no-tail diagnostic for the highest frequencies range. We are aware that numerous methods exist to attain a reliable estimation of wave parameters and we understand that researchers may apply their own method, in this case, they may ask for the raw data files.*

**Reviewer #2:**

The knowledge of lagoon scale hydrodynamics (either from observations or modelling) was essentially limited by poor observations. Especially, during tropical cyclone seasons, strong atmospheric events give rise to strong effects to the changes of coastal and ecosystem processes. This article of "Hydrodynamic and hydrological processes within a variety of coral reef lagoons: Field observations during 6 cyclonic seasons in New Caledonia" highlights at the observations collected during the 6 cyclonic seasons in New Caledonia. Although these datasets are very locally, but in a long term, it will support the requirement to evaluate the climate change on the coastal marine environment. I think it is worthy of publishing this work at ESSD if they can improve or show more significance compare the previous background as the suggestions listed:

Comment #1 of Reviewer #2 (website):

In the Section 6, the overview about the data set describes the high-resolution sections for temperature, salinity and so on. I think this part should be improved more at the current version. Firstly, about the observation errors in these five regions or in the lagoons, could you add more analysis? Secondly, compared to other reference data or climatology, could you show the difference? It will highlight the values of these new observations. Finally, as the title indicated there are related to the 6 cyclonic seasons, could you conclude how far the distance (or the time lag) from the cyclones will lead to the clear impact on the lagoon hydrology? Any one of them will be give more scientific contribution.

Answer to Comment #1 of Reviewer #2: *We thank Reviewer #2 for this constructive comment. With regards to the first comment, we admit we are not sure to what the reviewer refers to with the term "errors". If the reviewer refers to observational errors due to instrument lags and faults, we can say that we have paid great attention to the systematic maintenance/calibration of all our instruments at frequencies recommended by the manufacturers (we have stated within the document, see Line 376). Furthermore, we have identified and documented any potential error-data points within the metadata of all datasets, such that data users are made aware of this before use and can adjust accordingly.*

*With regards to the reviewer's suggestion to compare our datasets with existing or past data, we remind the reviewer that, as mentioned in the manuscript (lines 204-205) we are working in unmonitored lagoons, which unfortunately means there were no data sets or climatologies to compare with. Furthermore, our intentions with this paper are to make available these first reference observations for future studies.*

*To finally address the last part of the comment, the aim of this data-paper was mainly to present an overview of the observations acquired and not specifically to analyze the impact of cyclones. A PhD work is on the way to analyze this specific point about cyclones and we thus decided not to spoil his work. Within this study, in section 6, we illustrated some examples of key signatures of cyclones that occurred during our sampling period, such as sea levels, sea-states, temperature, current, salinity on the forereef as well as inside lagoons and passes.*

Comment #2 of Reviewer #2 (website): As we known, around New Caledonia there are long history of observations around lagoons as said in Line 54-59. Are all the observation surveys included during the cyclonic seasons around NC, especially between 2014-2021? What are the relationships with the previous surveys? It looks the zone E has a bit overlap with the one at near Nouméa. So could you add more words to explain why to ignore those observations if existing?

Answer to Comment #2 of Reviewer #2: *As we mentioned in line 58-63, past research works have been centered into 2 lagoons (Nouméa during the period of 1995-2010 and Ouano around 2013-2015), and these datasets do not overlap with cyclone seasons or major cyclones. To our knowledge there are no others observational strategies targeting cyclonic season during our period of observations (2014-2021). Concerning past observations into Nouméa lagoon (Zone E), we ignore these past observations mainly because data are not accessible and sampling strategies were not dedicated to high-frequency measures during cyclonic seasons.*

Comment #3 of Reviewer #2 (website): As they said at Line 460: "All data sets presented herein are freely available on SEANOE in dedicated repositories in NetCDF format." However, based on my checking, the data links for SEARSE and SAR (https://doi.org/10.12770/dad19639-c901-4edb-85cd-1fd546aa4cdb, and http://dx.doi.org/10.12770/96e4f2ef-e809-4005-b5df-529adc4e3306) show they are not open access like the rest. So could you clarify it explicitly?

Answer to Comment #3 of Reviewer #2: *We thank Reviewer #2 for pointing this out and admit an oversight in how we presented this within the paper. We have made available the SEARSE and SAR data sets within the Sextant repository and we have clarified this within the text (lines 458-459) and include the working DOI links in Table 1.*

Comment #4 of Reviewer #2 (website): Figure 1 the letters from A-E at the top panel for the surveys are not followed chronologically as the color bar shown, which will be better for a good consistence.

Answer to Comment #4 of Reviewer #2: *We thank the reviewer for pointing this out and agree that it may cause confusion. We followed the comment and have changed the letters in Figure 1 and 2 accordingly to keep the chronological aspect. Moreover, we added these letters in the survey timeline in Figure 1.*

Comment #5 of Reviewer #2 (website): Some variables contained in the NC files can be improved for completed information. For example, one CTD profile in Dec. 2014 from SPHYNX surveys named 55275.nc provide very good field, but only one issue found about PAR which unit is not clearly stated.

float PAR(depth, station) ;

   PAR:_FillValue = -999.f ;

   PAR:longname = "Irradiance" ;

   PAR:units = "PAR/Irradiance, Biospherical/Licor" ;

Answer to Comment #5 of Reviewer #2: *We thank Reviewer #2 for highlighted this major error in CTD Netcdf files. We took this into account and have changed the units within the attributes of PAR variable to* "microEinsteins/meter^2sec" *and the long_name attribute to* "Irradiance Biospherical/Licor" *into each CTD files of SPHYNX, NOUMEA, CADHYAK and NEMO campaigns. We have reprocessed and reviewed all files and variables again, and have made these available in SEANOE database.*

Comment #6 of Reviewer #2 (website): Line 22: delete "stake" because the processes presented in this study like temperature and salinity variability also covering the natural variability.

Answer to Comment #6 of Reviewer #2: *We thank you for pointing this out, and would like to also state that we had presented our manuscript to two English native speakers, who have extensively reviewed, corrected and made recommendations to the language and grammar in this paper.*

Comment #7 of Reviewer #2 (website): Line 185: missing understanding for "the N/O Alis"

Answer to Comment #7 of Reviewer #2: *We thank Reviewer #2 for this comment, and understand the potential for confusion in using N/O, the French derivative for R/V (i.e., Research vessel). We modified as follows: R/V Alis in line 194.*

**Reviewer #3:**

The paper describes the measurement and data collection of different hydrodynamic parameters in different coral reef lagoons of New Caledonia. Collected during 6 tropical cyclone seasons, the data offers to improve our understanding of coral reefs in face of climate change. I think the paper can be published after a rework of the current version. Especially the data part is quite short compared to rest of the paper.

Comment #1 of Reviewer #3 (website): An extensive proofreading is needed. citations in text Split figures for easier understanding and better presentation, Section 8?

Answer to Comment #1 of Reviewer #3: *Manuscript has been extensively proofread by 2 natives E.S who enhanced the quality of English sentence (~50 % of the sentences rewritten/modified). We also added our two kinds English reviewers in the acknowledgement section. We agree with this comment and we split Figure 1 into 2 separate figures. References to figures are now modified within 3.2 section that describe lagoons morphologies and sampling strategies. We thank Reviewer #3 for pointed out the missing section 8 and we corrected it on line 476, the conclusion refers to the section 8 instead of 9.*

Comment #2 of Reviewer #3 (website): The description of the used instruments should be expanded. This should include manufacturers, instruments, sensors and special settings. e.g. Line 328: which instruments had a burst mode of 10/15 min

Answer to Comment #2 of Reviewer #3: *We take into accompt this comment and we create an extensive table available in Appendice section (Table A1) presenting the main information (type, dates, position, parameters, measurement timing) of all deployments in a given place. We also referenced the table in line 309.*

Comment #3 of Reviewer #3 (website): Same for data processing and show us more of the data e.g. one sample from each lagoon. Are there any (older) other data sets for comparison?

Answer to Comment #3 of Reviewer #3: *We decided to expose only few of our observation to give an idea to reader what kind of process we can extract from our dataset. As we mentioned in line 58-63, past research works have been centered into 2 lagoons (Nouméa during the period of 1995-2010 and Ouano around 2013-2015), and these datasets do not overlap with cyclone seasons or major cyclones. To our knowledge there are no others observational strategies targeting cyclonic season during our period of observations (2014-2021).*

Comment #4 of Reviewer #3 (website): Links to data on Ifremer server are difficult/impossible to access for non-French speaker. If possible add an English information text.

Answer to Comment #4 of Reviewer #3: *We thank Reviewer #3 for this relevant comment and we have translated the abstract presenting SEASER and SAR data sets in Sextant metadata pages and we added a sentence to help non-French speaker to download or visualize datasets.*